# Multiple pathways for glucose phosphate transport and utilization support growth of *Cryptosporidium parvum*

Rui Xu [1,3], Wandy L. Beatty[1], Valentin Greigert[1], William H. Witola [2] & L. David Sibley [1] ✉

*Cryptosporidium parvum* is an obligate intracellular parasite with a highly reduced mitochondrion that lacks the tricarboxylic acid cycle and the ability to generate ATP, making the parasite reliant on glycolysis. Genetic ablation experiments demonstrated that neither of the two putative glucose transporters CpGT1 and CpGT2 were essential for growth. Surprisingly, hexokinase was also dispensable for parasite growth while the downstream enzyme aldolase was required, suggesting the parasite has an alternative way of obtaining phosphorylated hexose. Complementation studies in *E. coli* support a role for direct transport of glucose-6-phosphate from the host cell by the parasite transporters CpGT1 and CpGT2, thus bypassing a requirement for hexokinase. Additionally, the parasite obtains phosphorylated glucose from amylopectin stores that are released by the action of the essential enzyme glycogen phosphorylase. Collectively, these findings reveal that *C. parvum* relies on multiple pathways to obtain phosphorylated glucose both for glycolysis and to restore carbohydrate reserves.

The apicomplexan parasite *Cryptosporidium* causes diarrheal disease in humans and animals[1]. The majority of infections in humans are caused by two species: *C. parvum*, which infects animals and causes zoonotic infection in humans, and *C. hominis*, which is primarily transmitted among humans[2,3]. *Cryptosporidiosis* is the second leading cause of severe diarrheal disease and death in children under one year old among resource-poor settings[4]. Transmission of cryptosporidiosis occurs through a fecal-oral route and the entire life cycle takes place in a single host. After sporozoites are released from the oocyst in the small intestine, they invaginate the host membrane to form a parasitophorous vacuole that is intracellular but extra-cytoplasmic. During intracellular growth, a highly invaginated membrane structure named the feeder organelle forms at the parasite-host interface[5,6]. The feeder organelle is unique to *Cryptosporidium* and it is comprised of convoluted membrane layers that form at the host-parasite interface and extend upward into the cytosol of the parasite[5,6]. This elaborate membrane system is believed to increase the surface area to facilitate nutrient uptake from the host[7]. Previous studies suggested that an ATP-binding cassette (ABC) transporter was localized to this interface based on wide field microscopy of antibody-stained samples[8], although the composition and functions of proteins at this interface are largely unknown.

*Cryptosporidium* has a reduced genome and greatly streamlined metabolic pathways and therefore is thought to depend to a greater extent on nutrient salvage[9–11]. For example, *Cryptosporidium* has a reduced mitochondrion-like organelle named the mitosome[12,13] and lacks many de novo synthesis pathways for amino acids, nucleosides, and fatty acids[14]. In addition, *C. parvum* and *C. hominis* lack the tricarboxylic acid (TCA) cycle, electron transport chain, and functional ATP synthase[14–16]. However, *Cryptosporidium* encodes all the genes in the glycolysis pathway, which is thought to provide the main source of energy for the parasite[11,17]. Consistent with this model, transcriptional

[1]Department of Molecular Microbiology, Washington University School of Medicine, St. Louis, MO 63130, USA. [2]Department of Pathobiology, University of Illinois Urbana-Champaign, Urbana, IL 61802, USA. [3]Present address: College of Veterinary Medicine, South China Agricultural University, Guangzhou 510642, China. ✉e-mail: sibley@wustl.edu

profiling suggests that glucose and amylopectin are important carbon sources for energy during fertilization and oocyst formation[18].

Apicomplexan parasites obtain glucose from the host by transport across their cell membrane via dedicated glucose transporters[19]. Glucose transporters belong to the major facilitator superfamily (MFS) and mediate the import of glucose, mannose, and fructose. *Cryptosporidium* and *Plasmodium* spp. contain two glucose transporters each, while *Toxoplasma gondii* contains five glucose transporter family members[19]. The *Plasmodium falciparum* hexose transporter (PfHT1) localizes to the plasma membrane during asexual stage and transports glucose and fructose[20]. Because it has an essential role in asexual stages of growth, PfHT1 has been proposed as a potential target for antimalarial drugs[21]. The *Toxoplasma gondii* sugar transporter TgGT1 is also localized to the parasite plasma membrane along with TgST2, while TgST1 and TgST3 localize to intracellular vesicles in tachyzoites[22]. TgGT1 is not essential for parasite survival because *Toxoplasma* can utilize glutamine as a carbon source through the TCA cycle to generate ATP[23]. It also has a pathway for gluconeogenesis to generate sugars needed for other functions[24]. However, *C. parvum* doesn't contain the TCA cycle and also lacks gluconeogenesis, suggesting that glucose transporters may be required for parasite growth.

In this study, we localized two putative *C. parvum* glucose transporters CpGT1 and CpGT2 to the feeder organelle. We utilized live imaging and proximity-based biotin labeling to further characterize other transporters at the feeder organelle. Surprisingly, we found *C. parvum* hexokinase was dispensable, and yet downstream enzymes in glycolysis are essential. Our studies show that the parasite has alternative pathways for obtaining hexose phosphates including direct transport by CpGT1 and CpGT2 and sequential cleavage of glucose-1-phosphate (G1P) from amylopectin through the action of glucan phosphorylase. Collectively, our findings reveal diverse pathways for hexose metabolism that support intracellular growth of *C. parvum*.

## Results

### *Cryptosporidium parvum* contains two putative glucose transporters localized to the feeder organelle

Two putative glucose transporters cgd3_4070 (referred to here as *CpGT1*) and cgd4_2870 (referred to here as *CpGT2*) were previously identified in the genome of *C. parvum* (*Cp*) based on ortholog clustering[19]. To gain a better understanding of their relationships, we compared the protein sequences of *Cp* glucose transporters (CpGT1, CpGT2) with the *P. falciparum* hexose transporter (PfHT1), the *T. gondii* glucose transporter (TgGT1), and several sugar transporters in *T. gondii* (TgST1, TgST2, and TgST3). Neighbor-joining phylogeny indicated CpGT1 and CpGT2 cluster together and they were closer to PfHT1 and TgGT1 (Fig. 1a). Pfam analysis showed CpGT1 and CpGT2 each contain 12 transmembrane domains with both the N-terminus and C-terminus predicted to be exposed on the cytoplasmic side of parasite membrane, thus matching a canonical glucose transporter structure (Fig. 1b). To explore the localization of these two transporters in *Cp*, we modified each gene using CRISPR-Cas9 to add three hemagglutinin epitopes (3HA) to the C-terminus of the target gene followed by a Nluc-P2A-Neo^R selection cassette. Following transfection and selection in mice, clonal lines with the proper genomic insertion were identified by diagnostic PCR (Supplementary Fig. 1a, b).

To assess the localization of epitope-tagged *Cp* proteins, extracellular sporozoites, and different stages of intracellular parasites were examined by immunofluorescence assay (IFA). Neither CpGT1-3HA and CpGT2-spaghetti monster HA (smHA) were detected in sporozoites but they were expressed in all intracellular life stages (Supplementary Fig. 2a, b). In trophozoites, CpGT1 and CpGT2 were localized within the perimeter of a prominent ring-like structure at the parasite-host interface that is defined by a previously described mAb 1B5[25] (Fig. 1c, d). In more advanced stages of asexual growth, CpGT1 and CpGT2 were found at the base of meronts as shown using confocal microscopy to generate Z-stacks that were rendered in 3D (Fig. 1c, d). Interestingly, there were subtle differences in the pattern with CpGT1 extending upward from the basal feeder organelle region, while CpGT2 was more heavily concentrated at the base (Fig. 1c, d), a difference that was seen in both immature meronts (prior to cytokinesis) and mature (after cytokinesis) meronts containing merozoites. By transmission immuno-electron microscopy (immuno-EM), CpGT1 and CpGT2 were detected on the convoluted membranes of the parasite feeder organelle that emanate from the host parasite interface but also along membranes that extended upward into the parasite (Fig. 1e, f). Wild type *Cp* used as a control for immuno-EM staining was consistently negative under similar conditions (Supplementary Fig. 3). Taken together, these results indicated that CpGT1 and CpGT2 were expressed in all intracellular life stages where they are localized to the *Cp* feeder organelle at the host parasite interface.

To examine the architecture of the membranes that separate the host and parasite, we also performed transmission EM of early growth stages of *C. parvum* using a previously described Air Liquid Interface (ALI) culture system[26]. The position of parasites growing at the apical surface of polarized epithelium in the ALI system preserves the orientation of the parasite, its surrounding membranes, and the host cell with greater fidelity than when cultured on non-polarized adenocarcinoma cells. Transverse sections through trophozoites growing in ALI cultures show clear demarcation of the outer host plasma membrane, the parasitophorous vacuole, and the parasite surface member (Fig. 1g). The feeder organelle is evident as a tortuous cluster of membranes at the base of the parasite that is subtended by a dense cytoskeletal meshwork in the host cell (Fig. 1g, h). Importantly, the parasitophorous vacuole membrane does not extend beneath the parasite but instead wraps back at a dense thickening and becomes contiguous with the parasite plasma membrane (Fig. 1h). As such, the feeder organelle is open to the host cytosol and presumably has access to nutrients through diffusion. Consistent with this model, CpGT1 and CpGT2 staining extended below the outline of plasma membrane, detected by mAb 1E12, to membranes of the feeder organelle that define the host parasite interface (Fig. 1c, d).

### The feeder organelle forms several hours after invasion and is enriched in transporters

Previous studies have suggested that the feeder organelle, which is a unique structure to *Cryptosporidium*, may be involved in import or export of small molecules[6]. Having found that putative *Cp* glucose transporters were expressed in the feeder organelle, we investigated the formation of this interface by time-lapse video microscopy. We engineered transgenic parasites expressing the mCherry fluorescent protein in parasite cytoplasm and a mNeon tag fused to the C-terminus of CpGT1 (Fig. 2a, Supplementary Fig. 1c). We infected HCT-8 cells with CpGT1-mNeon-mCh transgenic parasites and acquired multi-color epifluorescence images every 10 min from 30 min to 16 hours post infection (hpi), corresponding to a complete round of merogony. In a representative time-lapse series take from Supplementary Movie 1, mNeon expression was very low in sporozoites and the signal increased between 2 and 4 hpi, gradually becoming brighter with time (Fig. 2b). When quantified from multiple videos, the mNeon fluorescence gradually increased after 2 h reaching a peak at 8 h post invasion, when the parasite has finished two rounds of DNA replication and contains 4 nuclei (Fig. 2c). The mNeon signal decreased slightly from 8—12 hpi, as the parasite volume increased during the completion of merogony (Fig. 2c). The mNeon signal remained detectable at the host interface after merozoites were released from the parasitophorous vacuole, indicating the feeder organelle remains associated with the host cell after parasite egress (Fig. 2b). We also studied the formation of the feeder organelle by transmission EM from samples that were

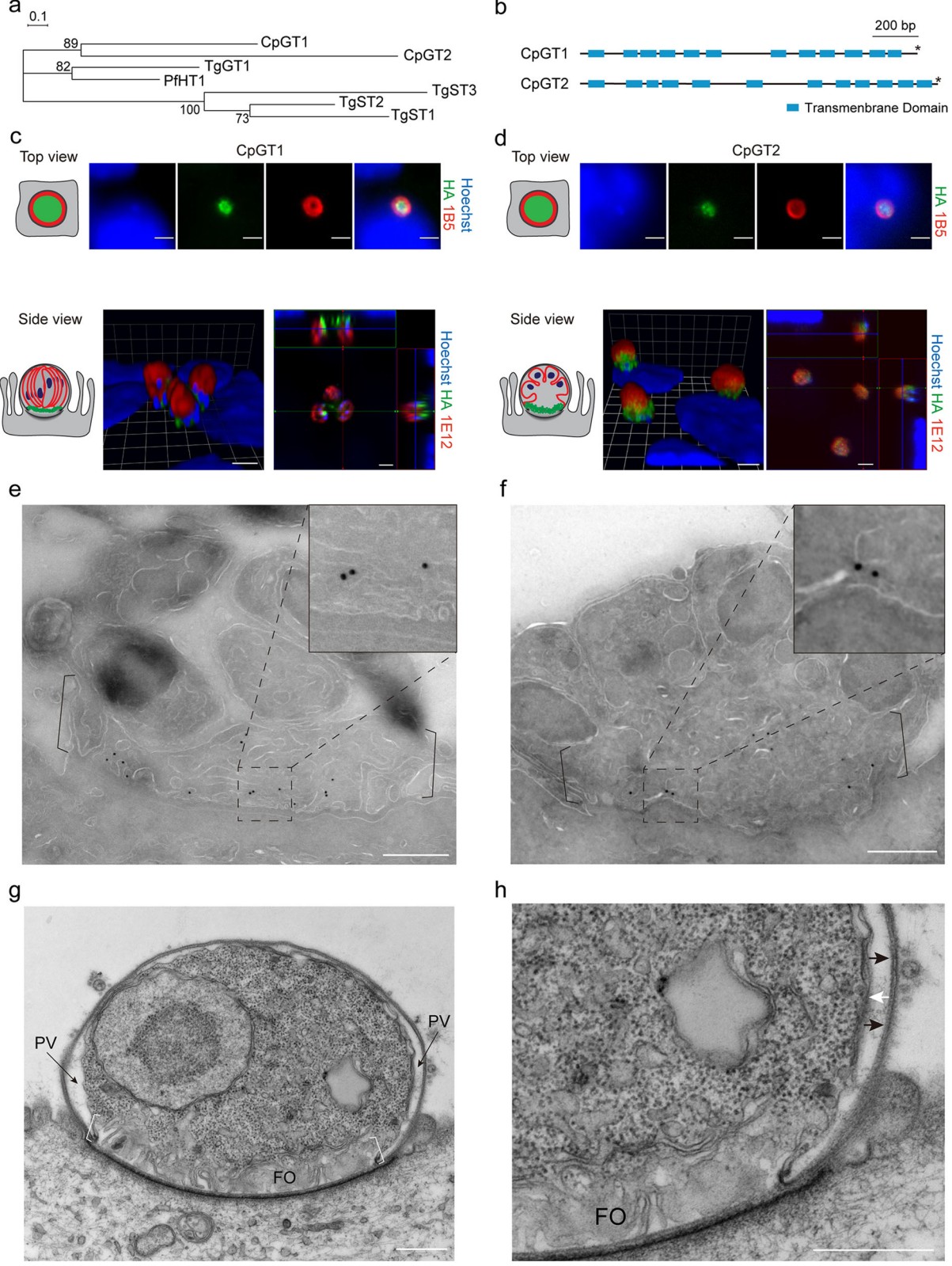

fixed at discrete time points after invasion. At early time points (i.e. 1–2 hpi), large, empty vacuoles were present at the host-parasite interface, but they lacked the convoluted membrane structure of the feeder organelle (Supplementary Fig. 4). At late time points (i.e. 3–4 hpi), multiple membrane layers were observed at the host-parasite interface (Supplementary Fig. 4). Collectively, these findings indicate that the feeder organelle is gradually assembled during the

first few hours after infection during which time CpGT1 and CpGT2 are expressed and trafficked to this membranous interface.

To identify additional proteins localized to the feeder organelle, we searched for proteins that are in close contact with CpGT1 using proximity-based biotin labeling[27]. For this purpose, we generated a transgenic parasite line with miniTurbo (mTurbo) and 3HA fused to the C-terminus of the *CpGT1* gene (Supplementary Fig. 1d). In brief, HCT-8

**Fig. 1 | *Cryptosporidium parvum* contains two putative glucose transporters CpGT1 and CpGT2 localized to the feeder organelle. a** Phylogenetic relationship of *C. parvum* (*Cp*) putative glucose transporters CpGT1 (gene ID: cgd3_4070), CpGT2 (gene ID: cgd4_2870), *Plasmodium falciparum* hexose transporter PfHT1, and *Toxoplasma gondii* glucose transporter TgGT1 and sugar transporter TgST1-3. The tree was constructed by a maximum likelihood method and JTT matrix-based model with 1,000 replications for bootstrapping. Scale bar = 0.1 of substitutions per amino acid site. **b** Schematic representation of the domain structure of CpGT1 and CpGT2 including putative transmembrane domain (blue) and stop codon (*). **c, d** Immunofluorescence localization of CpGT1-3HA (left) and CpGT2-smHA (right) in trophozoite (top view) and meront stages (side view) grown on HCT-8 cells. See also supplementary Fig. 1a. For trophozoite staining, cells were fixed at 4 hours post infection (hpi) and stained with rat anti-HA (green), mouse 1B5 (red) and Hoechst (blue). For meront staining, cells were fixed at 24 hpi and stained with rat anti-HA (green), mouse 1E12 that recognizes a surface membrane protein (red) and Hoechst (blue). The experiment was performed twice with similar results. Scale bars = 2 μm.

Cartoon images to the left of IFA panels illustrate plasma membrane (red) and transporters (green) in trophozoites (top, parasite nucleus has been omitted as it is out of the plane of focus) and mature meront (bottom left) and immature meront (bottom right). **e, f** Ultrastructural localization of CpGT1-3HA (left) and CpGT2-smHA (right). Parasites were grown in HCT-8 cells, fixed at 20 hpi and processed for immuno-EM and stained with rabbit anti-HA followed by 18-nm colloidal gold goat anti-rabbit IgG. Similar results were seen in multiple sections from one experiment. Scale bars = 500 nm. See also supplementary Fig. 3. **g** Transmission electron micrograph of *C. parvum* trophozoite showing the parasitophorous vacuole (PV) space and membranous feeder organelle (FO). Similar results were seen in multiple sections from one experiment. Scale bar = 500 nm. **h** Enlargement showing the organization of the parasitophorous vacuole membrane (black arrows) the parasite plasma membrane (white arrows) and the feeder organelle (FO). Mouse intestinal spheroids were cultured on transwells to create the air-liquid interface culture. Cells were infected with wild type parasites and monolayers were fixed and processed at 1 dpi. Scale bar = 500 nm.

cells were infected with CpGT1-mTurbo-3HA expressing parasites and biotin was added in the media followed by detection or purification with streptavidin (Fig. 2d). We confirmed by IFA that CpGT1-mTurbo-3HA was correctly localized, and after adding biotin in culture at 19 hpi for 1 h labeling, the cells were brightly stained with fluorescent streptavidin (Fig. 2e, Supplementary Fig. 5). Next, we affinity purified the biotinylated proteins on streptavidin beads and detected them by SDS-PAGE and immunoblotting. The addition of biotin resulting the labeling of numerous proteins in addition to several bands that were labeled even in the absence of added biotin (Fig. 2f). The most abundant proteins labeled in the absence of added biotin were host proteins that are endogenously biotinylated (i.e. acetyl-CoA carboxylase, pyruvate carboxylase and propionyl CoA carboxylase) (Supplementary Data 1). Contaminating *C. parvum* proteins (i.e. ribosomal and cytoskeletal proteins) were found in both samples containing biotin and those not treated with biotin, confirming that the same number of parasites were present in both samples (Supplementary Data 1). Samples from three independent immunoprecipitated experiments comparing biotin label to vehicle control (0.1% DMSO) were analyzed by liquid chromatography-tandem mass spectrometry (LC-MS/MS). To identify the most probable interactors of CpGT1, we filtered the candidates using the following criteria: (1) 95% peptide threshold, and 99% protein threshold, (2) at least 2 peptides in the average of 3 replicates, (3) statistical significance determined using unpaired Student's *t-test*, and (4) at least 2-fold enrichment in biotin versus in vehicle (Fig. 2g, Supplementary Data 1). Although the bait protein CpGT1 was detected in the dataset, it was not among the most abundant proteins, consistent with the observation that it was not recognized as a major band in the streptavidin blot (Fig. 2e), suggesting it might not be easily solubilized. Additionally, it is difficult to infer the abundance of actual interacting proteins from the frequency of detected peptides since biotin labeling and identification are influenced by protein size, proximity to the bait, available residues for conjugation, sensitivity to trypsin, and detection by MS. As such, we focused on pathways represented by enriched hits, rather than specific abundance. We identified 38 enriched proteins including CpGT1 and 18 additional proteins that were annotated as transporters in CryptoDB, as well as various components of the secretory pathways (e.g. HSP70, rab11, ARF-GAP, clathrin) (Fig. 2g). Among the transporters were 8 members belonging to a superfamily of ABC transporters, which have been reported to transport small molecules across the plasma membrane[28]. To confirm the predicted localization to the feeder organelle, we focused on CpABC1 (cgd1_700), which was previously shown to localize to the feeder organelle by wide field IFA[8]. Here, we localized CpABC1 by tagging the endogenous gene with 3HA using CRISPR-Cas9 (Supplementary Fig. 1e). When examined by IFA, CpABC1-3HA was expressed in all intracellular parasites except sporozoites, similar to the expression pattern of CpGT1 (Supplementary Fig. 6a).

Furthermore, CpABC1-3HA was precisely localized to the feeder organelle by laser scanning confocal microscopy and immuno-EM (Supplementary Fig. 6b, c). Collectively, these findings indicate that the feeder organelle is enriched in a variety of transporters that are predicted to import or export small molecules.

## Glucose transporters are individually dispensable for *C. parvum* growth

To explore the role of CpGT1 and CpGT2 in *Cp*, we attempted to generate transgenic knockout (KO) strains. CRISPR-Cas9 genome editing was used to replace the endogenous locus with mCherry driven by the *Cp* actin promoter followed by a Nluc-P2A-Neo^R selection cassette (Supplementary Fig. 1a, b). The transgenic parasites were isolated by selection in Ifng^−/− (GKO) mice treated with paromomycin drinking water[29], and the proper genomic insertion was confirmed by diagnostic PCR amplification (Supplementary Fig. 1a, b). Somewhat surprisingly, we were able to obtain viable KO of each strain, suggesting they are redundant. To evaluate the relative impact of the loss of *CpGT1* or *CpGT2* on parasite burden in vivo, we infected the *NOD scid gamma* (NSG) mice with the same numbers of epitope-tagged (TAG) or knockout (KO) strain parasites and monitored relative luciferase activity from fecal pellets collected from 3 days post-infection (dpi) to 30 dpi. When comparing the parasite burden between CpGT1 KO strain with CpGT1 TAG control strain, we found no difference in luminescence values between these two groups of mice (Fig. 3a). Additionally, the CpGT1 knockout line expanded with similar kinetics and peak values to lines bearing a deletion of other non-essential genes that have been made previously[30,31]. In contrast, we observed reduced parasite burden in mice challenged with the CpGT2 KO strain when compared with mice challenged with the CpGT2 TAG strain (Fig. 3b). Overall, these results revealed that CpGT1 is not essential for parasite growth, while CpGT2 is not essential but is required for optimal growth even when CpGT1 is present.

## Hexokinase is dispensable for growth and yet aldolase is essential

The *Cp* lacks a TCA cycle and doesn't have the capacity for gluconeogenesis, thus it is thought to rely exclusively on glycolysis as an energy source[11]. *C. parvum* lacks glucokinase, thus the only enzyme capable of phosphorylating glucose into glucose-6-phosphate is *Cp* hexokinase (CpHK, cgd6_3800) (Fig. 3c). Recombinant CpHK is a versatile enzyme capable of phosphorylating glucose, fructose, and mannose in vitro[32]. To examine the essentiality of the *CpHK* gene, we attempted to generate a KO using CRISPR-Cas9 genome editing (Fig. 3d). Surprisingly, we easily obtained this mutant and there was no attenuation when NSG mice were infected with the *CpHK* KO strain (Fig. 3e). PCR analysis confirmed the complete loss of the *CpHK* coding sequence in oocysts of the KO strain isolated from infected mice (Fig. 3f). Since CpHK was dispensable for *Cp*, we want to explore other

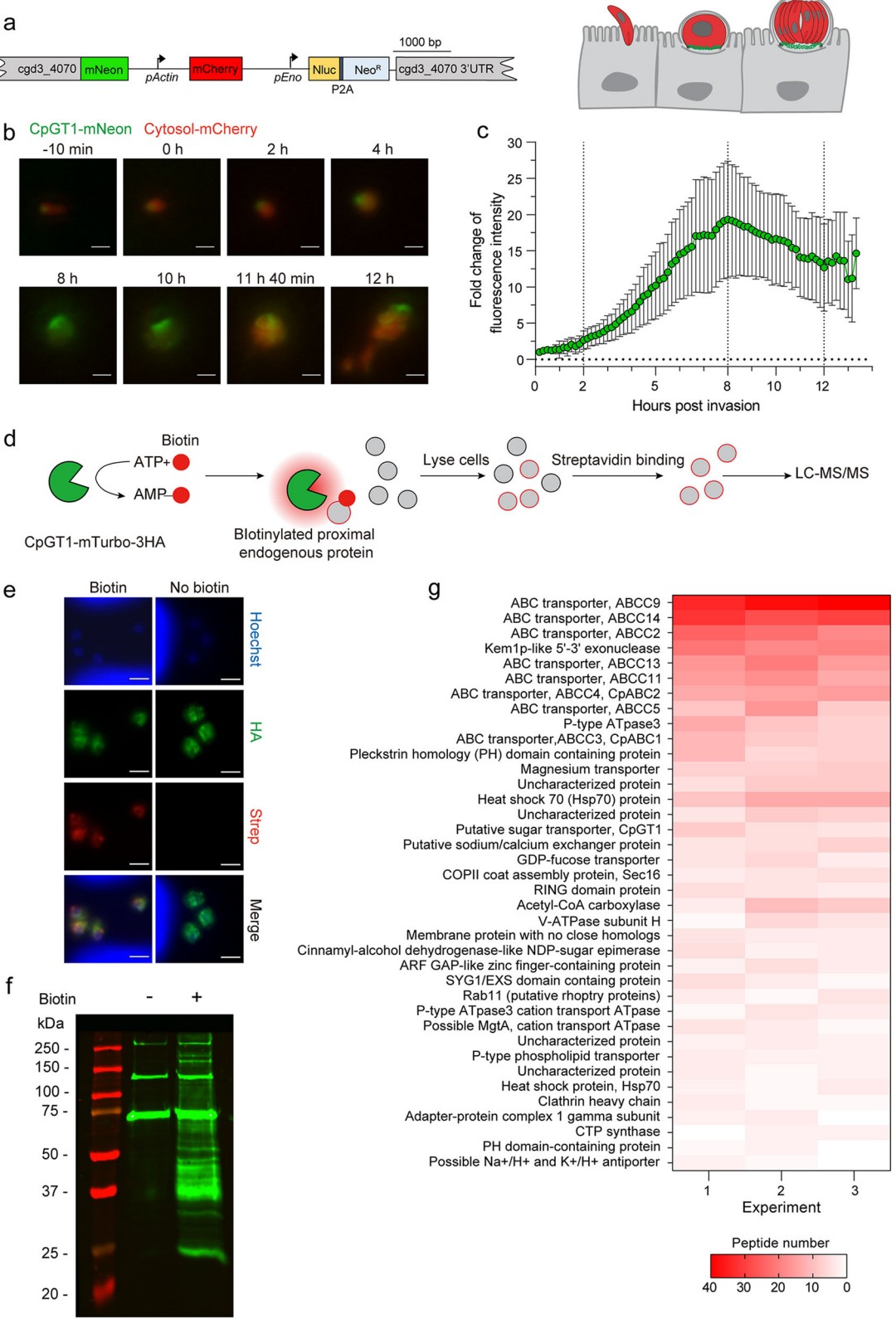

enzymes in glycolysis to determine if they were essential or not. We focused on *Cp aldolase* (cgd1_3020), which cleaves fructose-1,6-bisphosphate to form glyceraldehyde 3-phosphate and dihydroxyacetone phosphate. We attempted to disrupt the *Cp aldolase* gene using CRISPR/Cas9 (Supplementary Fig. 7a). However, despite multiple attempts, we failed to obtain a knockout aldolase in *Cp* (Supplementary Fig. 7b). Collectively these findings demonstrate that while

glycolysis is essential, hexokinase is not, implying that the parasite must have another source of phosphorylated hexose.

## CpGT1 and CpGT2 transport glucose-6-phosphate

Some bacteria have systems to directly salvage glucose-6-phosphate (G6P), including the *uhpT* gene that is the sole transporter for phosphorylated hexose in *Escherichia coli* (*E. coli*)[33,34] and the AfuABC

**Fig. 2 | Formation of the *C. parvum* feeder organelle and proximity labeling of enriched transporters. a** Schematic representation of CpGT1-mNeon-mCherry transgenic *Cp* strain. CpGT1 fused to mNeon and mCherry driven by *Cp* actin promoter. Cartoon depicts the labeling of the feeder organelle with CpGT1-mNeon and the cytosol with mCherry. **b** Time lapse microscopy at intervals during the merogony cycle. Scale bars = 2 μm. See also Supplementary Movie 1 for the time-lapse series of a full merogony cycle. **c** Fold change in fluorescent intensity of mNeon at time intervals during merogony normalized to 0 h post invasion. Each bar represents the mean ± SD for a total of 15 parasites from three combined experiments. **d** Schematic representation of proximity-dependent biotinylation in *Cp*. HCT-8 cells were infected with CpGT1-mTurbo-3HA parasites, labeled with biotin, lysed and affinity purified using streptavidin beads, and captured proteins identified by LC-MS/MS. See also Supplementary Fig. 1d for constructing and identification of CpGT1-mTurbo-3HA transgenic parasites. **e** Immunofluorescence of biotin-labeled parasites grown in HCT-8 cells. Cells were infected with CpGT1-mTurbo-3HA parasites. After 19 h post infection, cells were treated with 500 μM

biotin or vehicle (DMSO) for 1 h, then fixed and stained with rat anti-HA (green), streptavidin-Alexa 568 (red) and Hoechst (blue). The experiment was performed twice with similar outcomes. Scale bars = 2 μm. See also Supplementary Fig. 5 for immunofluorescence of biotin-labeled wild type parasite with anti-HA and strep-tavidin staining as a negative control. **f** Western blot analysis of biotinylated proteins in *Cp* cell lysates. Cells were infected with CpGT1-mTurbo-3HA parasites. After 19 h, cells were treated with 500 μM biotin or vehicle (DMSO) for 1 h, lysed in 1% NP-40 lysis buffer, captured on streptavidin beads and detected by Western blot with IRDye 800CW-labeled streptavidin (green). The experiment was performed three times with similar results. **g** Heat map of interactors of CpGT1 identified by mass spectrometry. Data were from three independent experiments including proteins (≥ 2 peptides, 95% peptide threshold, 99% protein threshold) with significant and 2-fold enrichment in biotin versus in vehicle ($P < 0.05$, unpaired Student's *t* tests, two-tailed). The numbers of peptides in biotin samples from three experiments were shown on the heat map. The list of gene ID is provided in Supplementary Data 1. Source data are provided as an accompanying Source Data file.

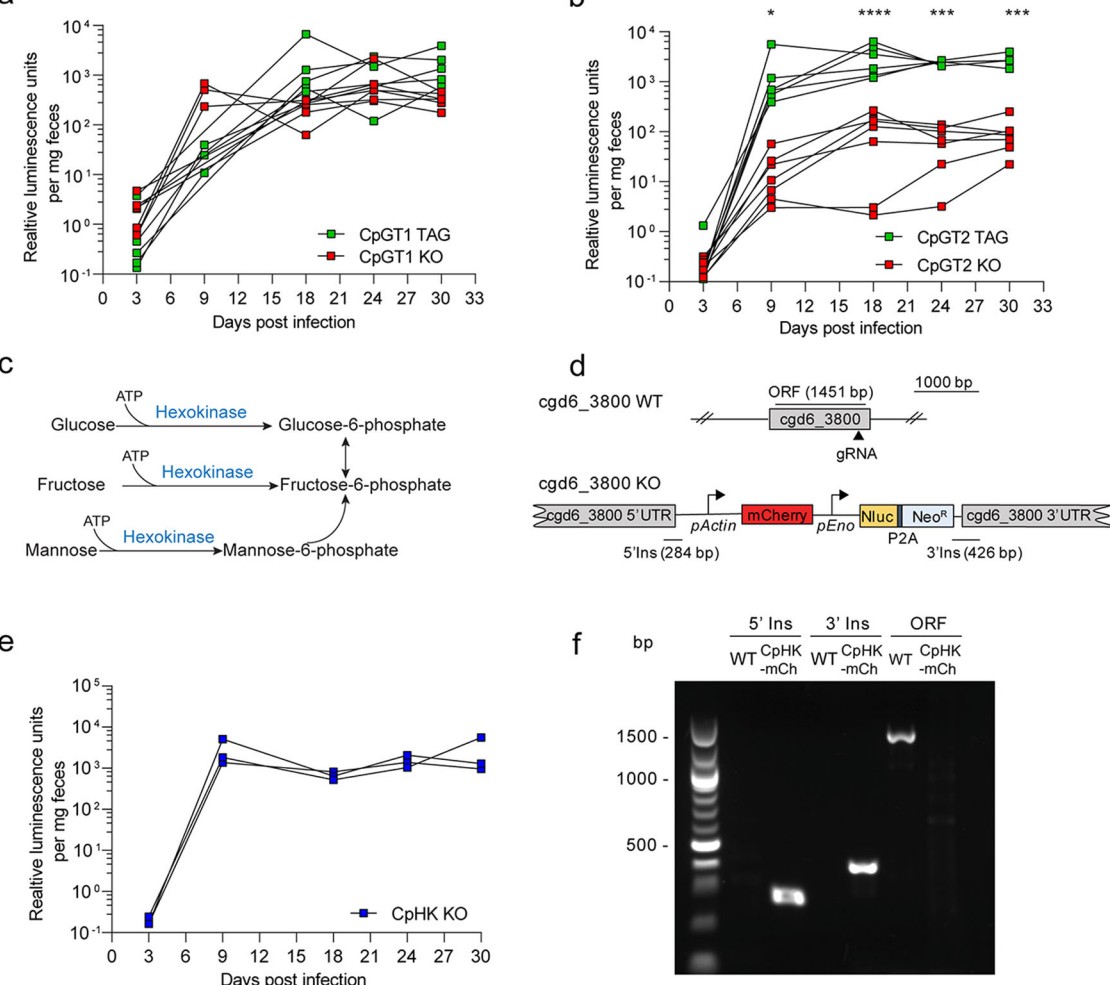

**Fig. 3 | Glucose transport and hexokinase are individually dispensable for *C. parvum* growth in vivo. a** Growth of CpGT1 tagging (TAG) or CpGT1 knockout (KO) transgenic strains **b** Growth of CpGT2 CpGT2 TAG or CpGT2 KO transgenic strains. NSG mice were infected with parasites and infection was monitored by measuring luciferase activity from fecal pellets collected at intervals post infection. Each line represents an individual NSG mouse (*n* = 6 for CpGT1 TAG, CpGT1 KO and CpGT2 TAG, *n* = 7 for CpGT2 KO, from two combined experiments). Two-way ANOVA corrected for multiple comparisons by Sidak's method (*, $P = 0.0250$, ***, $P = 0.0002$ at 24 hpi, ***, $P = 0.0001$ at 30 hpi, ****, $P < 0.0001$). No significant difference in parasite burden was observed between CpGT1 TAG and CpGT1 KO

strains. **c** Schematic representation of the reaction of hexokinase in the glycolytic pathway. **d** Diagram of the strategy to construct *Cp* hexokinase (HK) KO transgenic strain. **e** Growth of the CpHK KO transgenic strain. NSG mice were infected with parasites and infection was monitored by measuring luciferase activity from fecal pellets collected at intervals post infection. Each line represents an individual NSG mouse (n = 3 from one experiment). **f** PCR analysis of oocysts obtained from NSG mice infected with parasites as shown. Amplification products correspond to regions annotated in **d**. The experiment was performed twice with similar results. Source data are provided as an accompanying Source Data file.

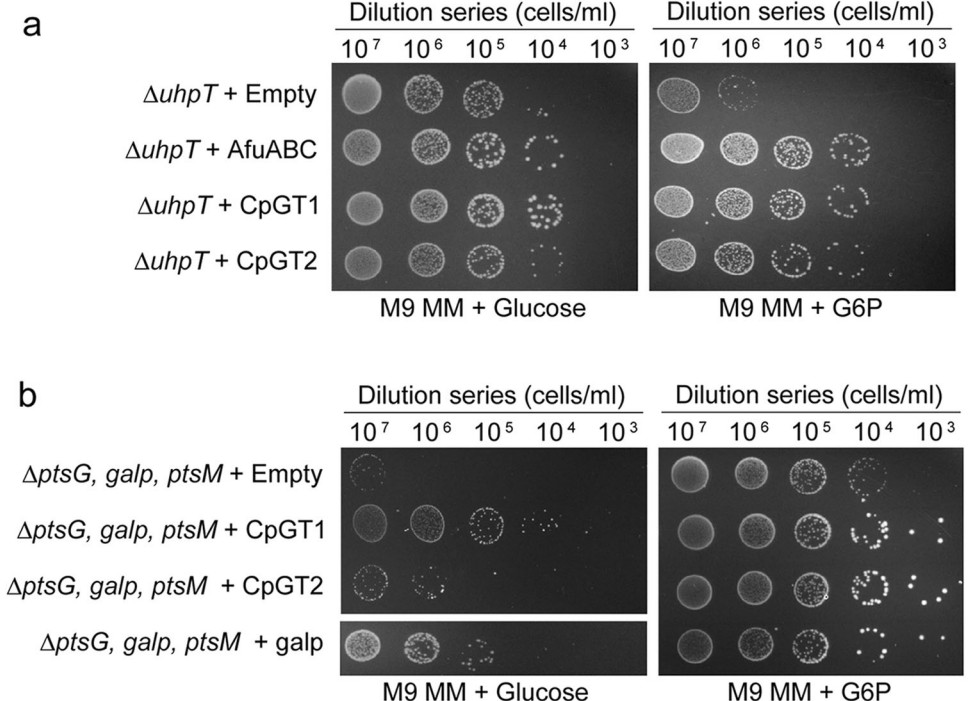

**Fig. 4 | The *C. parvum* CpGT1 and CpGT2 transporters salvage glucose phosphate. a** Solid media complementation of a mutant *E. coli* line lacking the glucose phosphate transport (*ΔuhpT*). Cells were grown in M9 minimal media (M9 MM) supplemented with 10 mM glucose or glucose-6-phosphate (G6P). The indicated dilutions were plated and grown at 37 °C for 40 h before imaging. The experiment was performed three times with similar results. **b** Solid media complementation of the *E. coli* the glucose transport mutant (*ΔptsG, Δgalp, ΔptsM*) grown in M9 MM supplemented with 10 mM glucose or G6P. The indicated dilutions were plated and grown at 37 °C for 40 h before imaging. The experiment was performed three times with similar results. See also Supplementary Fig. 8 for the growth assay of WT *E. coli* complementing with different genes. Source data are provided as an accompanying Source Data file.

sugar-phosphate transporter in *Actinobacillus pleuropneumoniae*[34]. Based on this precedent, we investigated whether CpGT1 or CpGT2 could transport glucose phosphate. Since *Cp* parasites are not amenable to direct transport assays, we used a genetic complementation system to test function. We complemented CpGT1 or CpGT2 in *E. coli* cells lacking uhpT[33,34]. As expected, the *ΔuhpT* mutant was unable to replicate on plates where G6P was the only carbon source (Fig. 4a). In contrast, when the mutant was transformed with a vector carrying *CpGT1* or *CpGT2*, we observed a significant increase in growth rate on M9 minimal medium plates supplemented with 10 mM G6P (Fig. 4a). The rescue observed with *CpGT1* and *CpGT2* was comparable to the rescue seen with AfuABC, which served as a positive control (Fig. 4a). We next sought whether *CpGT1* and *CpGT2* could also transport glucose. We used a genetic approach based on a triple mutant of *E. coli* lacking the *ΔptsG*, *Δgalp*, and *ΔptsM* glucose transporters that were generated using the lambda red system[35]. The *ΔptsG, Δgalp, ΔptsM* triple mutant transformed with an empty vector or a vector carrying *CpGT2* was unable to grow in the M9 minimal medium plates supplemented with 10 mM glucose (Fig. 4b). However, when the triple mutant was transformed with *CpGT1*, we observed modest growth on M9 minimal medium plates supplemented with 10 mM glucose compared to the empty vector-transformed cells, although the growth was not as strong as the *galp* positive control (Fig. 4b). We also validated that the growth rate differences were not due to inadvertent toxicity as there was no growth difference when they were expressed in K-12 *E. coli* (Supplementary Fig. 8). Overall, these findings predict that CpGT1 and CpGT2 can transport G6P, and CpGT1 can also transport glucose, in *E. coli*, suggesting they may have similar functions in *C. parvum*.

**Glycogen phosphorylase can provide phosphorylated hexose**
Given the non-essential nature of CpHK, we wondered if there were other mechanisms by which the parasite might obtain phosphorylated

hexose. *C. parvum* sporozoites contain amylopectin[17] and the genome contains glucan phosphorylase (GP, cgd6_2450) that debranches amylopectin into glucose-1-phosphate (G1P) (Fig. 5a). This substrate can be converted to G6P by phosphoglucomutase, a gene for which is also present in the *Cp* genome (Fig. 5a). We attempted to disrupt *CpGP* using CRISPR/Cas9 strategy, but were unsuccessful (Supplementary Fig. 7c, d). Based on evidence that GP was likely essential, we sought to develop a conditional knockout system in *Cp*. We implemented the auxin-based system for conditional depletion of CpGP protein in *Cp*, similar to that previously described for *T. gondii*[36]. We designed a targeting vector to fuse miniAID-3HA at the C-terminus of CpGP followed by codon-optimized auxin receptor TIR1 with a triple TY tag (TIR1-3TY) that was driven by the *Cp* actin promoter followed by a Nluc-P2A-Neo^R selection cassette (Fig. 5b, c). Following co-transfection with a CRISPR/Cas9 targeting plasmid, we isolated a stable transgenic *Cp* strain and verified the correct genomic insert by diagnostic PCR (Supplementary Fig. 1f). IFA analysis verified that CpGP-mAID-3HA and TIR1-3TY proteins were stably expressed in the parasite cytoplasm (Fig. 5d). We next validated the efficiency of knockdown in this system following treatment with indole acetic acid (IAA). The CpGP-mAID-3HA protein was still detected when the culture was treated with IAA for 1 h. However, following the addition of IAA for 6 or 24 h, the signal for CpGP-mAID-3HA was fully depleted (Fig. 5d). We confirmed that IAA did not affect wild-type parasite growth or host cell viability under these conditions (Supplementary Fig. 9a, b). To study the function of CpGP in *Cp*, we utilized the mAID system and measured the growth of parasites after conditional depletion of CpGP. We compared the growth of wild-type vs CpGP-mAID-3HA-TIR1 transgenic parasites following treatment with IAA or vehicle treatment for 24 h or 48 h. Auxin-induced depletion of CpGP-mAID-3HA protein did not affect parasite growth at 24 hpi but significantly inhibited parasite growth at 48 hpi (Fig. 5e, Supplementary Fig. 9c). Taken together, these data reveal that

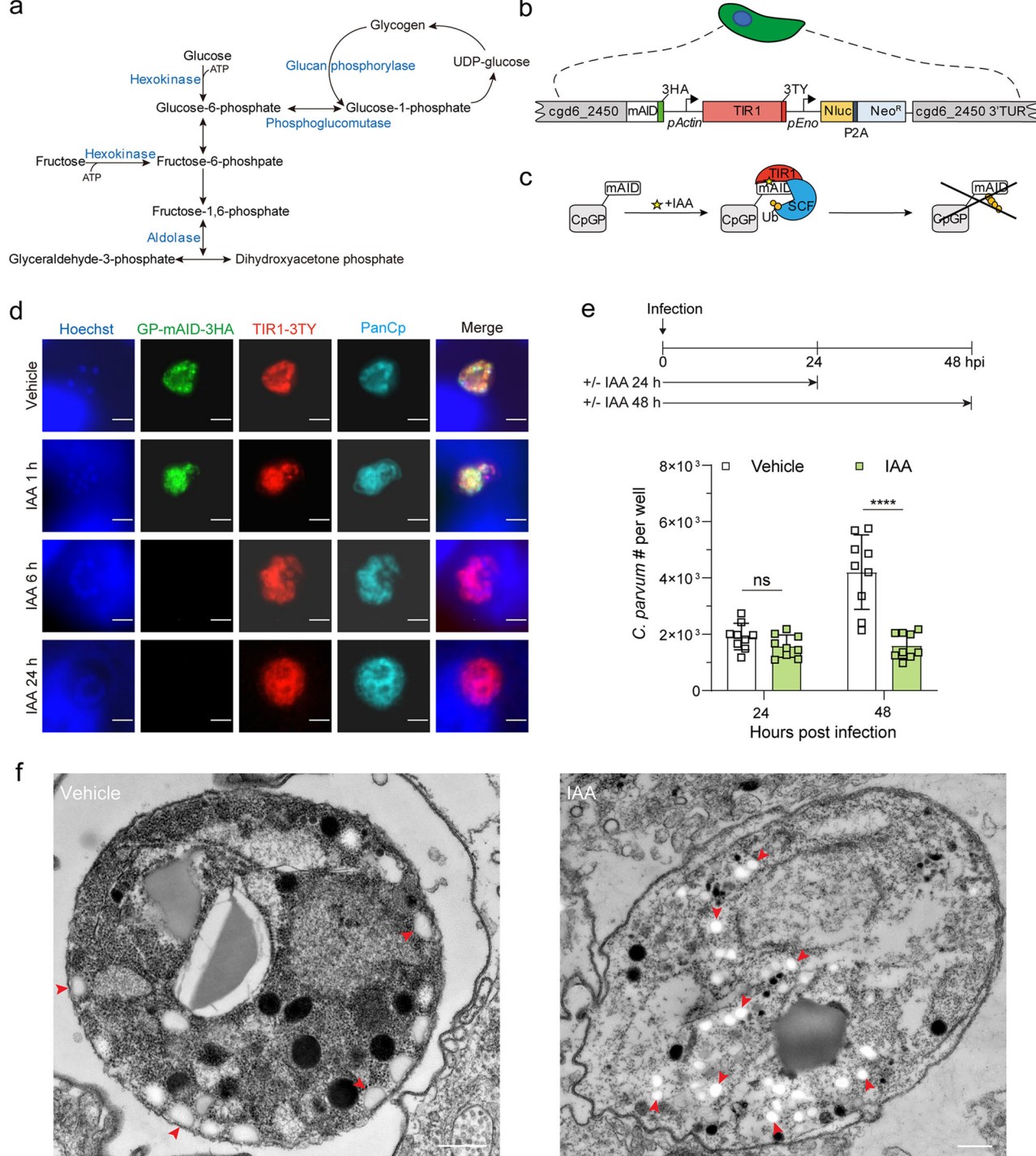

**Fig. 5 | *C. parvum* utilizes stored glycogen (amylopectin) for glucose phosphate as a carbon source. a** Schematic representation of stage I of glycolysis, glycogen breakdown and glycogen synthesis. **b** Schematic representation of *Cp* engineered to co-express the auxin receptor TIR1 from *Oryza sativa* and *Cp* glucan phosphorylase (CpGP) fused to mAID from *Arabidopsis thaliana*. See also Supplementary Fig. 1f for constructing and identification of CpGP-mAID-TIR1 transgenic parasites. **c** Schematic representation of conditional CpGP-mAID-TIR1 depletion. IAA, indole acetic acid; Ub, ubiquitin; SCF, Skp-1, Cullin, F-box (TIR1)-containing complex. **d** Immunofluorescence localization of CpGP-mAID-3HA expression. HCT-8 cells were infected with CpGP-mAID-TIR1 parasites and treated with 500 μM IAA or the vehicle (EtOH) for 24 h. Cells then were fixed and stained with rabbit anti-HA (green), mouse anti-TY (red), rat PanCp (cyan), and Hoechst (blue). The experiment was performed twice with similar outcomes. Scale bars = 2 μm. **e** Growth of CpGP-mAID-TIR1 parasites following treatment with IAA. HCT-8 cells were infected with

CpGP-mAID-TIR1 parasites, treated with 500 μM IAA or the vehicle (EtOH) for 24 h or 48 h. Cells were fixed, and labeled with rabbit PanCp and Hoechst, and the number of *Cp* in each well was determined by using a Cytation 3 imager. Each bar represents the mean ± SD for nine replicates in total from three experiments. Statistical analysis performed using was performed by two-way ANOVA corrected for multiple comparisons by Sidak's method. ns, not significant. ****, $P < 0.0001$. See also Supplementary Fig. 9c for the growth assay of host cell. **f** Transmission electron micrographs of CpGP-mAID-TIR1 parasites treated with vehicle (left) or IAA (right). HCT-8 cells were infected with CpGP-mAID-TIR1 parasites, and then treated with 500 μM IAA or the vehicle (EtOH). After 48 hpi, monolayers were fixed and processed for EM. Arrows (red) point to electron lucent amylopectin granules. Similar results were seen in multiple sections from one experiment. Scale bars = 500 nm. Source data are provided as an accompanying Source Data file.

CpGP is important for parasite growth and establish that the mAID system can be used to study essential proteins in *C. parvum*.

Previous studies in *T. gondii* have shown that disruption of GP causes alterations in the accumulation of amylopectin granules that are toxic to the parasite[37]. As such, we considered whether the loss of GP in *C. parvum* might impart a growth defect for similar reasons. Initially, we attempted to stain amylopectin using periodic acid Schiff reagent. However, the high concentration of O-linked glycans in the parasite resulted in a very high background. Instead, we used transmission EM to examine CpGP-mAID-3HA-TIR1 parasites grown in HCT-8 cells following treatment with IAA vs. vehicle control. Amylopectin granules are recognizable by their bright, electron-lucent staining, and have previously been described in oocysts and sporozoites of *C. parvum*[17] as well as in macrogamonts[38]. In control cells treated with vehicle, granules consistent with the properties of amylopectin were seen infrequently in macrogamonts (Fig. 5f). In contrast, CpGP-mAID-3HA-TIR1 cells treated with IAA contained numerous small, bright vesicles resembling amylopectin (Fig. 5f). These clusters of amylopectin-like vesicles occupied substantial area of the cytosol (Fig. 5f). Additionally, both the nucleus and cytosol were much less electron dense in IAA treated cells, suggesting the cells were adversely affected by the loss of GP and accumulation of amylopectin-like vesicles (Fig. 5f).

## Discussion

Although previous studies have emphasized the importance of glycolysis in *C. parvum* and *C. hominis*, the pathways by which hexose is acquired and metabolized have not been investigated. In this study, we characterized two glucose transporters, CpGT1 and CpGT2, which localized to the parasite feeder organelle. Gene deletion studies indicate that neither of them is essential for *Cp* growth in vivo. More surprisingly, genetic ablation studies revealed that hexokinase was dispensable, while aldolase was essential for parasite growth. Although these findings confirm that glycolysis is important, they also indicate an alternative pathway may provide phosphorylated glucose. Our findings suggest that CpGT1 and CpGT2 can directly transport glucose phosphate from the host cell, as they were able to complement *E. coli* mutants deficient in this ability. In addition, *Cryptosporidium* also has an alternative way to get glucose phosphate by debranching amylopectin through the action of glycogen phosphorylase, which was essential for *Cp* growth. Our findings indicate that *Cryptosporidium* utilizes multiple pathways to obtain phosphorylated hexoses necessary for glycolysis and to support intracellular development within the host.

The feeder organelle is an elaborate system of membranes that forms at the host-parasite interface where it is thought to facilitate nutrient uptake from the host[39,40]. However, knowledge of the formation, composition, and function of this organelle is limited. Here we used immuno-EM to localize CpGT1 and CpGT2 to the membranous interface comprising the feeder organelle. CpGT1 was expressed at very low levels in sporozoites, and time-lapse video microscopy of a fluorescent reporter strain indicated that CpGT1 gradually increased expression over the first few hours and accumulated at the feeder organelle. Although we have not verified the location of the endogenous proteins lacking tags (i.e. using antibodies to the protein), the transgenic lines expressing tagged proteins grew normally, suggesting the tags do not disrupt function. Parallel EM studies on the development of the feeder organelle showed that large, empty vacuoles were first detected at 2 hpi with the formation of membrane sheets by 3-4 hpi. These findings suggest that the feeder organelle is not formed at the time of invasion, but rather lags by several hours, during which time proteins destined for this interface are upregulated. Taking advantage of the position of CpGT1 at this interface, we used permissive biotin labeling to identify additional transporters in feeder organelle. Based on the MS data, the ABC transporter family was the largest

protein family expressed in the feeder organelle. This family of multi-membrane-spanning proteins has been implicated in transport a diverse range of substrates including amino acids, lipids, ions, and sugars[41]. The ABC transporters play a role in lipid metabolism in *Plasmodium falciparum*[42] and are implicated in multidrug resistance[43,44]. *C. parvum* contains 21 members of the ABC transporter family[28], including CpABC1 which was previously shown to be present at host-parasite interface by wide-field immunofluorescence microscopy[8]. Here, we confirmed that CpABC1 was localized to the feeder organelle by laser scanning confocal and immuno-EM microscopy. Together, these results confirm that the feeder organelle contains a variety of putative transporters and predict this interface is critical for exchanging small molecules between the parasite and host. Our identification of several proteins that are translocated to the feeder organelle, as well as defining the kinetics for its assembly, will support future studies to define the origin and composition of the membranes and features that direct proteins to this interface. Additionally, future studies are likely to build on these findings through alternative extraction techniques to improve the efficiency of recovery of membrane proteins and additional controls to rule out contaminants.

In contrast to *Toxoplasma* and *Plasmodium*, *C. parvum* and *C. hominis* have compact genomes that encode a highly streamlined metabolism[14], lacking the TCA cycle, gluconeogenesis, and cytochrome-based respiratory chain[10,11]. Hence, glycolysis is thought to provide the sole source of endogenously synthesized ATP for *C. parvum* and *C. hominis*[45]. Glucose transporters identified in *Toxoplasma* and *Plasmodium* are important for optimal parasite growth, and these organisms can also salvage glutamine and shunt it through the TCA cycle to generate ATP[21,22]. Here, we found that genetic ablation of CpGT1 did not alter parasite growth, and loss of CpGT2 resulted in only mild attenuation of oocyst shedding. These findings suggest that these two transporters are partially redundant, a prediction that could be tested using regulated knockdown together with gene disruption to generate double mutants. *Cryptosporidium* contains a single hexokinase that catalyzes the first reaction in glycolysis. Surprisingly, knockout of *CpHK* did not affect parasite growth while we were unable to disrupt *Cp* aldolase, indicating that glycolysis is likely essential and suggesting that there is an alternative source for phosphorylated hexoses. Consistent with this model, we demonstrate that CpGT1 and CpGT2 are capable of salvaging phosphorylated hexoses when expressed in *E. coli*, suggesting they may also do so in *C. parvum*. However, our studies do not completely rule out other interpretations; for example, *C. parvum* may harbor a cryptic sugar kinase capable of phosphorylating hexoses, thus bypassing the need for HK. Additionally, the apparent essentiality of aldolase could be due to difficulty in targeting this locus, a possibility that could be tested by regulated knockdown. Alternatively, the inability to disrupt aldolase may be related to the accumulation of toxic intermediates in its absence, as was previously shown in *T. gondii*[46].

Using a complementation assay in *E. coli*, we demonstrated that CpGT2 can support growth on glucose phosphate while CpGT1 also supported growth on glucose phosphate and to a lesser extent, on glucose. We cannot be certain that the activities inferred by the rescue of growth in *E. coli* fully reflect the contributions of CpGT1 and CpGT2 in the parasite since heterologous complementation may be incomplete due to differences in expression, protein folding, or regulation. However, direct measurement of transport activities in *C. parvum* is complicated by the presence of much larger host cells, and since the transporters studied here are not expressed in sporozoites, they remain inaccessible to current monitoring techniques. Previous studies in prokaryotic organisms have shown that *E. coli* can uptake phosphorylated hexose using the uhpT transporter[47]. There are also other sugar phosphate transporters in Gram-negative bacteria such as AfuABC in *Actinobacillus pleuropneumoniae*[34] and UhpABCT in *Vibrio cholerae*[48]. The presence of hexose phosphate transporters in bacteria

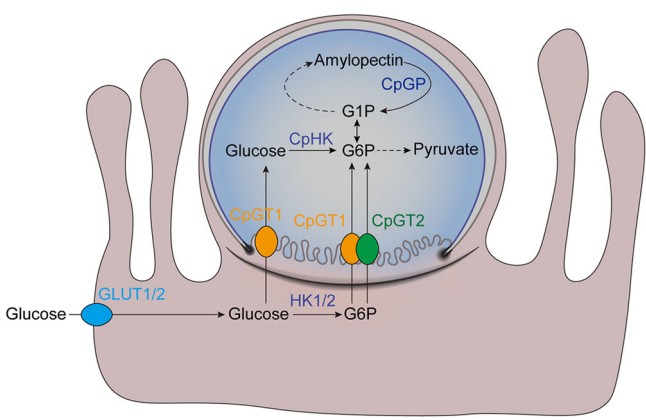

**Fig. 6 | Schematic representation of transportation and utilization of glucose phosphate in *C. parvum*.** *C. parvum* has multiple pathways to transport and utilize hexoses for generating energy. Glucose from the gut lumen can be transported into epithelial cells through the action of host GLUT1 or GLUT2 receptors, then phosphorylated to G6P by HK1 or HK2. Glucose in the host cells could be transported into *Cp* by CpGT1 and G6P could be transported into *Cp* by CpGT1 and CpGT2 at the feeder organelle. The parasitophorous vacuole does not extend beneath the parasite but curves upward at an electron dense boundary to meet the parasite plasma membrane (see EM examples in Fig. 1). Once in the parasite cytosol, glucose can be phosphorylated by hexokinase (CpHK) and used for glycolysis or converted to G1P and used for amylopectin synthesis. GLUT1/2, human glucose transporter 1 or 2; HK1/2, human hexokinase 1 or 2. Image generated with assistance from Abigail Kimball.

may reflect their specialized niche in the intestine where hexose phosphates are present[34] due to production by the host microbiota[49]. However, to our knowledge, glucose phosphate transporters have not been reported previously in eukaryotes. *C. parvum* and *C. hominis* are encased within a membrane at the apex of the host cell, and they likely obtain these nutrients from the host cell via transport at the feeder organelle. Furthermore, calves infected with *C. parvum* exhibit a decrease in the plasma glucose levels and an increase in expression of hexokinase 2 leading to more glucose phosphate in epithelial cells[50]. In combination, these results suggest that *C. parvum* and *C. hominis* directly transport hexose phosphates from the host cell cytosol to support optimal growth. Additionally, *C. parvum* expresses numerous genes annotated as nucleotide sugar transporters[51], and it is possible that some members of the encoded proteins are present at the feeder organelle to obtain modified sugars from the host.

*Cryptosporidium* also has an alternative way to produce glucose phosphate through the degradation of amylopectin by glucan phosphorylase that generates glucose-1-phosphate, which in turn is converted to glucose-6-phosphate by phosphoglucomutase. Amylopectin is used for carbon storage and has been shown to play biological roles in many apicomplexan parasites[52,53]. Amylopectin granules have been detected in the *Cp* oocysts and sporozoites by transmission electron microscopy[17] and their abundance correlates with sporozoite viability[54]. However, the roles of amylopectin and the regulatory mechanisms that control its turnover during the parasite infection remain to be elucidated. In this study, we failed to directly knockout CpGP in vivo suggesting it plays a crucial role in *Cryptosporidium*. Consistent with this prediction, conditional degradation of CpGP using the mAID degron system showed a significant decrease of parasite growth at 48 hpi in vitro. The reason for this delayed timing is uncertain but it suggests that initially the intracellular replicating stages acquire simple sugars from the host cell, and only later require additional energy stores during the emergence of merozoites, or during the development of sexual stages, which occurs after 36 hpi. Alternatively, previous studies found that deletion of GP in *Toxoplasma* caused an accumulation of amylopectin and reduced replication rates

of tachyzoites in vitro and parasite virulence in vivo[37,55]. Consistent with this finding, we also observed an increase in electron lucent amylopectin-like vesicles in macrogamonts of *C. parvum* following degradation of GP. Although we have not been able to verify the biochemical composition of these vesicles due to technical limitations, they are highly suggestive that the accumulation of amylopectin granules in macrogamonts impairs parasite growth. Hence, CpGP appears to be required not only for providing substrates for glycolysis but also for maintaining the proper balance of amylopectin synthesis and digestion. To restore this cycle, the parasite requires glycogen synthase and a de novo source of G-1-P to reform amylopectin (Fig. 6). Our findings indicate phosphorylated hexose can be derived either from hexokinase activity on transported glucose or by direct import of G-6-P by CpGT1 and CpGT2 (Fig. 6). Importantly, the architecture of membranes at the host-parasite interface predicts that the feeder organelle is exposed directly to the host cytosol, thus providing access to small molecules by diffusion. Collectively, these findings reveal metabolic flexibility in *C. parvum* and *C. hominis* for utilization of glucose, which likely reflects a strategy to maximize environmental sources of energy in light of their highly streamlined genome and greatly reduced mitochondrion.

## Methods

### Animal studies and ethical approval
Animal studies on mice were approved by the Institutional Animal Studies Committee (School of Medicine, Washington University in St. Louis). Ifng[-/-] mice (referred to as GKO) (002287; Jackson Laboratories), and *Nod scid gamma* mice (referred to as NSG) (005557; Jackson Laboratories) were bred in-house in a specific-pathogen-free animal facility on a 12:12 light-dark cycle. Male and female mice between 8 and 10 weeks of age were used to perform experiments. Mice were co-housed with siblings of the same sex throughout the experiments. Animals that became non-ambulatory during the course of infection were humanely euthanized as specified in the protocol.

### Parasite culture
*Cryptosporidium parvum* isolate AUCP-1 was maintained by the repeated passage in male Holstein calves and purified from fecal material[56]. All calf procedures were approved by the Institutional Animal Care and Use Committee (IACUC) at the University of Illinois at Urbana-Champaign. Purified oocysts were stored at 4 °C in phosphate-buffered saline (PBS) plus 50 mM Tris and 10 mM EDTA (pH 7.2) for up to six months after fecal collection. Before infection of cell culture or animal models, *Cp* oocysts were treated in a 40% bleach solution (containing 8.25% sodium hypochlorite) diluted in Dulbecco's phosphate-buffered saline (DPBS; Corning) for 10 mins on ice, then washed three times in DPBS containing 1% bovine serum albumin (BSA; Sigma). Bleached oocysts were stored for up to 1 week in DPBS plus 1% BSA at 4 °C before infection. For most experiments, cells were infected with bleached oocysts added directly to the cell culture medium. For experiments requiring sporozoites, bleached oocysts were excysted in 0.75% sodium taurocholate diluted in DPBS at 37 °C for 1 h.

### Cell culture
Human ileocecal adenocarcinoma cells (HCT-8; ATCC CCL-244) were cultured in RPMI 1640 ATCC Modification medium (Thermo Fisher A14091-01) supplemented with 10% fetal bovine serum (Gibco) at 37 °C in a 5% $CO_2$ incubator. Cells were confirmed to be mycoplasma-free with the e-Myco plus *Mycoplasma* PCR detection kit (Boca Scientific).

### Bacteria culture
*Escherichia coli* strain DH5α was used for cloning. Wild type *E. coli* K-12 (BW25113), Δ*uhpT* (JW3641), and Δ*ptsG* strains (JW1087) were obtained from the Keio collection of single gene knockouts[57]. Bacteria were

grown at 37 °C on Luria-Bertani (LB) agar plates or in LB broth with appropriate antibiotics: kanamycin 50 µg/ml, ampicillin 100 µg/ml.

## Phylogenetic analysis

The amino acid sequences of CpGT1 (cgd3_4070) and CpGT2 (cgd4_2870) in *C. parvum*, *Plasmodium falciparum* hexose transporter PfHT1 (PF3D7_0204700), *Toxoplasma gondii* glucose transporter TgGT1 (TGME49_214320) and sugar transporters TgST1-3 (TGME49_257120, TGME49_272500, TGME49_201260) were extracted from VEuPathDB (https://veupathdb.org/veupathdb/app/). MUSCLE was used to align the concatenated sequences and phylogenetic trees based on maximum likelihood were constructed with 1000 replications for bootstrapping.

## Primers

All primers were synthesized by Integrated DNA Technologies and are listed in Supplementary Data 2 in the supplemental material.

## Construction of CRISPR/Cas9 plasmids for genetic modification of *C. parvum*

Homology repair templates and CRISPR/Cas9 plasmids were generated as previously described[29,58]. CRISPR/Cas9 plasmids were generated by adding a sgRNA targeting the gene of interest into the linear Cas9 plasmid that was amplified from pACT:Cas9-GFP, U6:sgRNA (addgene plasmid no. 128552) via Gibson assembly cloning. For targeting the CpGP gene for knockout, a double sgRNA strategy was used to target regions at the 5' and 3' ends of the gene[29]. Primers for the following constructs are described in Supplementary Data 2.

**Epitope tagging and fluorescence reporters.** For tagging of genes with the HA epitope, we modified the previously described pINS1-3HA-Nluc-P2A-neo[29] to replace the INS1 C-terminal and 3' UTR sequences with corresponding regions from the gene of interest using a four-fragment Gibson assembly (New England Biosciences)[59]. For tagging CpGT2 with the spaghetti monster HA tag (smHA), the tag was amplified from a previously described vector pLIC-SM-HA (addgene plasmid no. 111188) and assembled in a five fragment Gibson assembly. For generation of the CpGT1-mNeon-mCh plasmid for live imaging, the following segments were connected using Gibson assembly: the mNeon gene amplified from pTubLinker-TYx3-mNeon-TYx3-HX (addgene no. 166237), the mCherry tag driven by the *Cp* actin promoter amplified from pUPRT-mCh-Nluc-P2A-neo-UPRT[58], and the plasmid backbone amplified from the p4070-3HA-Nluc-P2A-neo.

**Permissive biotin labeling.** For permissive biotin labeling, miniTurbo[27] was added as a C-terminal fusion to CpGT1. The sequence of miniTurbo was codon optimized for *Cp* by Azenta Life Sciences and synthesized by Integrated DNA Technologies. The miniTurbo domain was then inserted into p4070-3HA-Nluc-P2A-neo using Gibson assembly to generate the plasmid p4070-miniTurbo-3HA-Nluc-P2A-neo.

**Knockouts and conditional degradation.** To generate knockouts, homology repair fragments were PCR amplified from pUPRT-mCh-Nluc-P2A-neo-UPRT with primers that also contained 50 bp 5'UTR and 3'UTR homology regions for the genes interest.

For conditional degradation of CpGP, we generated a plasmid to add the auxin-induced degradation domain (miniAID) with 3HA epitope tag to the C-terminus of GP. The plasmid also contained the Tir1 gene tagged with 3TY and driven by the *Cp* actin promoter to facilitate degradation[60]. Gibson assembly was used to connect the following fragments: the plasmid backbone amplified from pINS1-3HA-Nluc-P2A-neo, Nluc-P2A-neo[R] amplified from pUPRT-mCh-Nluc-P2A-neo-UPRT, *Cp* actin promoter from pUPRT-mCh-Nluc-P2A-neo-UPRT, TIR1 that was codon optimized for *Cp* by Azenta Life Sciences and synthesized by Integrated DNA Technologies, 3TY tag amplified from pTubLinker-TYx3-mNeon-TYx3-HX plasmid, miniAID-3HA sequence amplified

from pTUB1:YFP-mAID-3HA,DHFR-TS:HXGPRT (addgene plasmid no. 87259)[36] and C-terminal and 3' UTR homology regions amplified from *Cp* genomic DNA.

## Generation and amplification of transgenic parasites

The $1 \times 10^8$ sporozoites were resuspended in SF buffer (Lonza) containing 100 µg of tagging or knockout plasmids, or 50 µg amplicon DNA, together with 50 µg CRISPR/Cas9 plasmid in a total 200 µl volume. The mixtures were then transferred to a 100-µl cuvette (Lonza) and electroporated on an AMAXA 4D-Nucleofector system (Lonza) using program EH100. Electroporated sporozoites were transferred to cold DPBS and kept on ice before infection. To select transgenic parasites, GKO mice were orally given 8% (wt/vol) sodium bicarbonate 5 min prior to infection and then orally gavaged with electroporated sporozoites. Transgenic parasites were selected with 16 g/l paromomycin (Biosynth International, Inc) provided in the drinking water starting at 24 h and continuing ad lib. To amplify transgenic parasites or monitor oocysts burden, NSG mice were administered the fecal slurry containing $2 \times 10^4$ oocysts from an infected GKO mouse, and paromomycin drinking water was used for selection immediately after infection. Fecal pellets were collected after 3 dpi and stored at −80 °C for DNA extraction or at 4 °C for luciferase assay or for purification.

## Luciferase assay

To monitor parasite shedding, luciferase assays were performed on fecal materials using Nano-Glo luciferase assay kit (Promega) as previously described[29]. Relative luminescence units per milligram of feces were calculated using the average luminescence reading of two technical replicates per sample divided by the mass of the fecal pellets.

## PCR genotyping

DNA was extracted from fecal pellets using the QiaAmp PowerFecal pro-DNA kit (Qiagen). To confirm proper template integration for each transgenic strain, PCR analysis was used to amplify the genomic regions. PCR was performed using purified fecal DNA template, PrimeSTAR GXL Premix (Takara), and primers listed in Supplementary Data 2. PCR gels were viewed on the ChemiDoc MP imaging system (Bio-Rad) and imaged using Image Lab v6.1 (Bio-lab).

## Immunofluorescence microscopy

For imaging extracellular parasites, sporozoites were added to coverslips coated with poly-L-lysine (Advanced Biomatrix), fixed with 2% formaldehyde for 10 min and permeabilized and blocked with TSS buffer (DPBS containing 1% BSA and 0.1% Triton X-100) for 20 min. For imaging intracellular parasites, HCT-8 cells were plated on 12-mm-diameter glass coverslip (Thermo Fisher Scientific) in 24-well tissue culture plates and incubated until confluent. Monolayers were infected with oocysts and incubated 4 h for trophozoite staining or 24 h for meront staining and 48 h for microgamont and macrogamont staining. Cells were fixed with 4% formaldehyde for 10 min, washed twice with DPBS and then permeabilized and blocked with TSS buffer for 20 min. Primary antibodies were diluted in blocking buffer for staining: rat anti-HA (Roche) was used at 1:500, 1B5 and 1E12 (purified mouse mAbs) was used at 1:500, 4D8 (hybridoma supernatant) was used at 1:20, and PanCp (rabbit pAb) was used at 1:2000. Cells were incubated with primary antibodies for 60 min at room temperature, washed in PBS, and Alexa fluor-conjugated secondary antibodies (Thermo Fisher Scientific) diluted 1:1000 in blocking buffer for 60 min. Finally, nuclear DNA was stained with Hoechst 33342 (5 µg/ml, Thermo Fisher Scientific) for 15 min and then mounted with Prolong Diamond Antifade Mountant (Thermo Fisher Scientific). Images were captured on a Zeiss Axioskop Mot Plus fluorescence microscope equipped with a 100×, 1.4 N.A. Zeiss Plan Apochromat oil objective lens or on a Zeiss LSM880

laser scanning confocal microscope equipped with a 63×, 1.4 N.A. Zeiss Plan Apochromat oil objective lens. Images were acquired using AxioVision Rel v 4.8, software or ZEN v2.1, v2.5 software. Images were manipulated in ImageJ v2.0.0 (https://fiji.sc/) or rendered in 3 dimensions using Volocity v6.3 software. For images captured from Hoechst staining, the intensity was electronically increased (i.e. adjusted white level) similarly in all panels to better reveal the nuclei of the parasite.

## Immuno-electron microscopy

HCT-8 cells were plated in 6-well culture plates and grown until confluent. Cells were infected with transgenic oocysts and incubated for 20 h. Monolayers were washed twice with DPBS and fixed in freshly prepared mixture of 4% paraformaldehyde and 0.05% glutaraldehyde (Polysciences Inc.) in 100 mM PIPES/0.5 mM $MgCl_2$ buffer (pH 7.2) for 60 min at 4 °C, then embedded in 10% gelatin and infiltrated overnight with 2.3 M sucrose/20% polyvinyl pyrrolidone in PIPES/$MgCl_2$ at 4 °C. Samples were trimmed, frozen in liquid nitrogen, and sectioned with a Leica Ultracut UCT7 cryo-ultramicrotome (Leica Microsystems Inc.). Ultrathin sections of 50 nm were blocked with 5% fetal bovine serum/5% normal goat serum for 30 min and subsequently incubated with rabbit anti-HA antibody (Sigma) diluted 1:1000 for 60 min at room temperature. Following washes in block buffer, sections were incubated with 18 nm colloidal gold conjugated goat anti-rabbit IgG (H + L) (Jackson ImmunoResearch Laboratories Inc.) diluted 1:20 for 60 min. Sections were stained with 0.3% uranyl acetate/2% methyl cellulose and viewed on a JEOL 1200 EX transmission electron microscope (JEOL USA Inc.) equipped with an AMT 8 megapixel digital camera and AMT Image Capture Engine v602 software (Advanced Microscopy Techniques). All labeling experiments were conducted in parallel with controls omitting the primary antibody.

## Transmission electron microscopy

Mouse or human intestinal spheroids were cultured on transwells to create the air-liquid interface culture using a modification of previously published methods[26,61]. Cells were infected with WT sporozoites. Monolayers were fixed at intervals post-infection in a freshly prepared mixture of 1% glutaraldehyde (Polysciences Inc.) and 1% osmium tetroxide (Polysciences Inc.) in 50 mM phosphate buffer for 30 min at 4 °C, then embedded in 3% agarose. Samples were rinsed in cold $dH_2O$ prior to staining with 1% aqueous uranyl acetate (Ted Pella Inc.) at 4 °C for 3 h. After several rinses in $dH_2O$, samples were dehydrated in a graded series of ethanol and embedded in Eponate 12 resin (Ted Pella Inc.). Sections of 95 nm were cut with a Leica Ultracut UCT ultramicrotome (Leica Microsystems Inc.), and stained with uranyl acetate and lead citrate. Samples were viewed and captured as described above.

## Live imaging

HCT-8 cells were plated in 24-well glass bottom plates (Cellvis) and grown until confluent. Cells were switched to prewarmed RPMI-1640 supplemented with 10% FBS without phenol red and infected with CpGT1-mNeon-mCh oocysts. Imaging was performed beginning at 30 min after infection and multiple points of interest were taken every 10 min for up to 16 h and captured using a Zeiss Observer Z1 inverted microscope with a Colibri LED illumination for multi-color epifluorescence. A Plan-Apochromat 63×, 1.4 N.A. oil objective lens, and an ORCA-ER digital camera were used for image acquisition. Growth conditions were maintained throughout each imaging experiment at 37 °C and 5% $CO_2$. Images were acquired and processed using ZEN 2.5 software. The fluorescence intensity of AF488 channel was manually measured from a region of interest of the same size across the time series. Fold changes of fluorescence intensity were normalized to 0 h post invasion. Data from three independent experiments were combined and analyzed using GraphPad Prism v9 software.

## Biotin labeling with miniTurbo

HCT-8 cells were plated at $2 \times 10^6$ cells per well in 6-well culture plates and grown for 24 h until confluent. Cells were infected with $2 \times 10^6$ CpGT1-miniTurbo-HA oocysts per well and cultured for 19 h. Then the infected monolayers were incubated with 500 µM biotin or vehicle control (0.1% DMSO) for 60 min. Labeling was stopped by transferring the cells to ice and washing them five times with cold DPBS. For indirect immunofluorescence microscopy, cells were fixed with 4% formaldehyde, permeabilized and blocked with TSS buffer, and labeled with rat anti-HA diluted 1:1000 in TSS buffer followed by anti-rabbit Alexa Fluor 488 diluted 1:1000, Streptavidin conjugate Alexa Fluor 568 (Thermos Fisher Scientific) diluted 1:1000 in TSS buffer, and Hoechst for nuclear stain. Images were captured and processed as described above. For immunoprecipitation, cells from one well of 6-well culture plate were lysed in 1% NP-40 lysis buffer (1% NP-40, 50 nM Tris HCl, 150 mM NaCl) supplemented with 1 × EDTA-free protease inhibitor cocktail (Roche) and Benzonase (Sigma) for 30 min on ice with gentle pipetting and then centrifuged at 14,000 × g for 15 min at 4 °C. The supernatant was loaded on the 0.5 mg streptavidin-conjugated magnetic beads (Thermo Scientific) and incubated overnight at 4 °C with gentle rotation. The beads were washed twice with 0.5% NP40 lysis buffer, once with 1 M KCl, once with 0.1 M $Na_2CO_3$, once with 2 M urea in 10 mM Tris HCl (pH 8.0), twice with 0.5% NP40 lysis buffer and finally washed five times with PBS. The bead samples were stored at −80 °C prior to Western blot analysis or LC-MS/MS processing.

## Western blot analysis

Samples were prepared in Laemmli buffer containing 100 mM dithiothreitol (DTT), boiled for 10 min, resolved by SDS-PAGE, and transferred to nitrocellulose membranes. The membranes were blocked with 3% fat-free milk in PBS and then probed with IRDye 800CW streptavidin (LI-COR Biosciences) diluted 1:2000 in PBS containing 0.1% Tween 20. Membranes were washed with PBS containing 0.1% Tween 20, and then scanning on Li-Cor Odyssey imaging system v 3.0 (LI-COR Biosciences).

## Mass spectrometry (MS) analysis

Samples from three independent biological replicates were submitted to the Nebraska Center for Biotechnology at University of Nebraska-Lincoln (Lincoln, NE, USA). Beads were reduced in 50 mM ammonium bicarbonate with 5 mM DTT for 1 h at 37 °C and then alkylated with 20 mM iodoacetamide for 30 min. The reaction was quenched with DTT and digested with trypsin overnight at 37 °C. The samples were dried, redissolved in 5% acetonitrile, 0.5% formic acid, and then run by nanoscale liquid chromatography (nanoLC)-MS/MS using 2 h gradient on a 0.075 mm × 250 mm Water CSH $C_{18}$ column feeding into a Thermo Eclipse mass spectrometer. Data were searched against the following proteomes using Mascot (v2.7.0, Matrix Science): *Cryptosporidium parvum* Iowa II (UniProt number UP000006726; 3805 sequences), *Homo sapiens* (UniProt number UP000005640; 20, 594 sequences), and common contaminants (cRAP 20150130; 124 sequences). Scaffold (v4.8.9, Proteome Software Inc.) was used to validate MS/MS based peptide and protein identifications. The validated data sets and filtered using a peptide threshold of 95%, a protein threshold of 99% and a minimum of 2 peptides. The fold change in peptide counts was determined using the ratio of total unique peptide counts in biotin labeling samples relative to no biotin samples averaged in three independent experiments. The average total unique peptide count was adjusted to 1 for proteins with no peptides detected in no biotin samples for fold enrichment calculation. Statistical analyses were performed in Scaffold using unpaired Student's *t*-tests ($P < 0.05$), two-tailed. For visualization, unique peptide counts in biotin labeling samples from three independent experiments were plotted in Prism v9 generating a heat map.

## Auxin-induced depletion of CpGP-mAID-HA protein

HCT-8 cells were cultured on coverslips as described above and infected with CpGP-mAID-TIR1 oocysts. Cells were treated with 500 µM IAA or vehicle control (0.1% ethanol) for 1 h (23–24 hpi), 6 h (18–24 hpi) or 24 h (0–24 hpi) or continuously treated with vehicle control for 24 h. Monolayers were fixed and stained at 24 hpi with rabbit anti-HA (Thermo Scientific) diluted 1:1000 followed by anti-rabbit Alexa Fluor 488 diluted 1:1000, mouse anti-TY (purified mouse mAbs) diluted 1:500 followed by anti-mouse Alexa Fluor 568 diluted 1:1000, rat PanCp diluted 1:2000 followed by anti-rat Alexa Fluor 647 diluted 1:1000 (artificially colored cyan in digital images) and Hoechst for nuclear stain. Images were captured and processed as described above.

## *C. parvum* growth assay with IAA treatment

HCT-8 cells were plated at $1 \times 10^5$ cells per well in black-sided, optically clear-bottomed 96-well plates (Greiner Bio-One) and grown until confluent. Cells were infected with $1 \times 10^5$ bleached wild type oocysts per well, and treated with 500 µM IAA or vehicle control for 48 h. After 48 hpi, monolayers were fixed in 4% formaldehyde for 10 min, washed twice in DPBS, and then permeabilized and blocked with TSS buffer for 20 mins. Parasites were labeled with rabbit PanCp diluted 1:2000 followed by anti-rabbit Alexa Fluor 488 diluted 1:1000. Host cell nuclei were stained with Hoechst for 20 min. Plates were imaged with a 10× objective on a BioTek Cytation 3 cell imager (9 images per well in a 3 × 3 grid). For measurements of CpGP-mAID-TIR1 parasite growth with or without IAA treatment, HCT-8 cells were infected with $5 \times 10^4$ bleached CpGP-mAID-TIR1 oocysts per well. Monolayers were treated with 500 µM IAA or vehicle control for 24 h or 48 h. After 24 hpi or 48 hpi, monolayers were fixed and stained as described above. Plates were imaged with a 20× objective on a BioTek Cytation 3 cell imager (36 images per well in a 6 × 6 grid). BioTek Gen5 software v3.08 (Agilent) was used to quantify the total number of parasites (puncta in the GFP channel) and host cells (nuclei in the DAPI channel) per well.

## Depletion of glucose transporters in *E. coli*

The *ΔptsG* strain (JW1087) of *E. coli* was used for deleting the genes *galp* and *ptsM* as previously described[35]. Plasmids pKD4, pKD46, and pCP20 were used for the construction of gene deletion mutants. The plasmid pKD46 was transfected into *ΔptsG* in order to express lambda red recombinase. The kanamycin resistance cassette with flanking homologous regions specific for *galp* or *ptsM* were amplified by PCR from the pKD4 plasmid. The resulting amplicons were transfected in to *ΔptsG* and selected on kanamycin plates. The kanamycin resistance cassette was eliminated by transformation with pCP20 plasmid and verified by PCR. Sequential rounds of this technique were used to get the triple knock out *ΔptsG*, *Δgalp*, and *ΔptsM*.

## Cloning and expression of CpGT1 and CpGT2 in *E. coli*

The sequences of CpGT1 or CpGT2 were codon optimized for *E. coli* and synthesized by Integrated DNA Technologies. The sequence of AfuABC (*Actinobacillus ferric* uptake ABC) gene was amplified from pSC101-AfuABC[34]. The sequence of *galp* gene was amplified from *E. coli* genome. Each of these fragments linked with 3HA epitope was inserted into the pSC101 vector[34] using Gibson assembly and assembled vectors transformed in parallel into three backgrounds: wild type *E. coli*, *ΔuhpT* mutant, and *ΔptsG*, *Δgalp*, *ΔptsM* mutant. Overnight cultures were grown in 5 ml of M9 minimal media (Sigma) with 10 mM glucose (Sigma) and 50 µg/ml kanamycin, 100 µg/ml ampicillin for mutants or 100 µg/ml ampicillin for wild type *E. coli*. Cells were washed twice with M9 minimal media (no carbon) and diluted to a density of $10^8$ cells/ml. For the growth assay, a dilution series of $10^7$ to $10^3$ cells/ml were plated in 5 µl spots on M9 minimal media plates with 10 mM glucose or 10 mM glucose-6-phosphate (Sigma) and antibiotics. Plates were incubated at

37 °C for 40 h, the plate with 10 mM glucose culturing triple mutant transformed empty vector, CpGT1 or CpGT2 vector were incubated at 37 °C for 50 h. Then, plates were imaged with ChemiDoc Imaging System (BioRad). The experiment was performed as three independent biological replicates.

## Statistical analyses

Statistical analyses were conducted in GraphPad Prism v9. Continuous variable data were analyzed for normality and appropriate non-parametric or parametric tests were applied based on the distribution of data. No data were excluded from analyses. Information on replicates, specific statistical tests, corrections for multiple comparisons, and *P* values are given in the legends for each figure.

## Reporting summary

Further information on research design is available in the Nature Portfolio Reporting Summary linked to this article.

## Data availability

All of the data are found in the main figures or supplementary information. Source data are provided with this paper.

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

## Acknowledgements

We thank Daniel Goldberg and members of the Sibley and Goldberg laboratories for helpful advice and Abigail Kimball for assistance with drawing the model. Imaging studies were conducted with assistance from the Molecular Microbiology Imaging Facility. Mass spectrometry was conducted by Sophie Alvarez and Michael Naldrett, Proteomic & Metabolomics Facility (RRID:SCR_021314), Nebraska Center for Biotechnology at the University of Nebraska-Lincoln. The pKD4, pKD46 and pCP20 plasmids were generously gifted by Anna Hooppaw and Mario Feldman in the Department of Molecular Microbiology, at Washington University in St. Louis. The pSC101-AfuABC plasmid was generously gifted by Trevor F. Moraes in the Department of Biochemistry at the University of Toronto. Supported by a grant from the National Institutes of Health (AI145496, LDS).

## Author contributions

Conceptualization, R.X., V.G., L.D.S.; Investigations, R.X., W.L.B., V.G., L.D.S.; Data curation, R.X., W.L.B., V.G.; Formal analysis R.X.; Methodology R.X., W.L.B.; Generation of research materials, R.X., W.L.B., W.H.W., Supervision, L.D.S.; Writing and Editing R.X., L.D.S., Revisions, all authors.

## Competing interests

The authors declare no competing interests.
