## [Peer Review File · Nature Communications]

Multiple pathways for glucose phosphate transport and utilization support growth of *Cryptosporidium parvum*REVIEWER COMMENTS

Reviewer #1 (Remarks to the Author):

Rui Xu et al. investigated how *C. parvum* obtains vital sugars from its host cells. They figured out two glucose transporters, CpGT1 and CpGT2 localized in the feeder organelle, were not essential for parasite growth through CRISPR-Cas9 strategy. In addition, they analyzed biotinylated proteins by CpGT1-miniTurbo-3HA to identify additional proteins localized to the feeder organelle. Complementation studies in *E. coli* demonstrated the two transporters could uptake G6P. Moreover, they suggested aldolase and glycogen phosphorylase as essential enzymes for parasite growth through CRISPR-Cas9 strategy.

In summary, the manuscript explores metabolic pathways of *C. parvum* with various techniques such as CRISPR-Cas9 knock out/knock in, proximity labeling, complementation studies in *E. coli*, and mAID degron system. Overall, the article seems to contribute to the understanding of *C. parvum* biology. The authors may find the specific points below useful to improve the manuscript.

Suggestions/questions:

1. In several experiments, the authors attached genetic tags such as HA, fluorescent protein, miniTurbo, mAID, etc., to specific proteins. The authors may need to show these genetic tags didn't interfere the specific proteins' functions.
2. In Figure 2e, no biotin condition shows streptavidin signal, while biotin condition doesn't. Please provide correct images.
3. In Figure 2e or 2f, please provide WT CpGT1 (without miniTurbo) +biotin control images.
4. In Figure 2f, please provide HA blotting to confirm the enzyme expression levels were same.
5. Please provide more details on proximity labeling experiment in the methods section. For example, how much cells/proteins were used for SA-beads enrichment? And how much SA-beads were used?
6. Have you analyzed biotinylated samples without streptavidin enrichment? Are the bands in - biotin condition endogenously biotinylated proteins in HCT-8 cells?
7. Could you clarify the reason for incubating the CpGT1-miniTurbo parasites for 19 hr?
8. In proteomics dataset, could you provide spectrum intensities, not only number of unique peptides?
9. To map non-restricted organelles, like the feeder organelle, it would be necessary to incorporate spatial controls, such as *C. parvum* cell membrane, for performing ratiometric analysis. The ratiometric analysis would enable the distinction of proteins that are universally abundant or easily biotinylated. For example, ABCC9 could've been labeled easily due to its big size.
10. As the authors have developed a conditional knockout system in *C. parvum*, is it possible to apply the same approach to aldolase? Failures of KO strain construction might not be enough evidence to define the enzyme as an essential one.

Reviewer #2 (Remarks to the Author):

In this manuscript, Xu et al. reports that the zoonotic protozoon *Cryptosporidium parvum* (Cp) possesses multiple transporters to acquire glucose or phosphorylated glucose from host cells to start the glycolysis. Significant discoveries include:

- 1) the investigators produced transgenic parasite lines that tagged two sugar transporters with

3HA, namely CpGT1 (cgd3_4070) and CpGT2 (cgd4_2870), and confirmed by immunolabeling and immuno-EM that both transporters were located to the “feeder organelle” during the parasite intracellular development, but absent in the motile sporozoites.

2) Since the unique “feeder organelle” has been poorly understood, they used a proximity-labeling technique to discover a number of new proteins potentially localized in this organelle (mostly transporters). They further validated that one of these proteins, CpACBP1, was localized in the feeder organelle.

3) CpGT1 and CpGT2, as well as the hexokinase (CpHK, cgd6_3800), were individually dispensable, indicating that Cp uses multiple pathways to start the glycolysis (some redundancy in acquisition of sugars). They also implied that glycolysis was essential based on the failure of knockout (KO) of aldolase gene in the parasite.

4) The glucan phosphorylase (CpGP, cgd6_2450) was essential (indispensable) for the parasite growth in vivo as implied by the failure of gene KO. Conditional KO suggested the CpGP plays a role in the parasite growth “in vitro”.

5) By transformation of *Escherichia coli* mutants deficient with glucose and glucose-6-phosphate (G6P) transporters with parasite transporters for complementary experiments, they showed that CpGT1 may transport both glucose and G6P, while CpGT2 may only transport G6P.

These discoveries are highly significant in understanding the parasite acquisition of carbohydrates from the host for energy metabolism and for carbon source, the function of feeder organelle unique to *Cryptosporidium* parasites. I have no major concerns, but a few minor comments as follows.

1) The starch (amylopectin) as a carbon storage could be essential for the environmental and motile stages, such as sporozoites in the oocysts and merozoites, but less likely essential to the intracellularly developing stages that could directly acquire mono- or di-sugars or phosphorylated sugars from the host. This may explain the “delayed” effect of CpGP-KO on the parasite growth in vitro (Fig. 5e), i.e., in 48-h in vitro assay, there is a release of merozoites to invade new host cells, by which CpGP-KO merozoites could only use stored mono/di-sugars and could not stay extracellularly for less time.

2) In *E. coli* complementary assays (Fig. 4), parasite CpGT1 and CpGT2 may not function optimally for many reasons. Therefore, the comparison on the efficiency of CpGT1 between transporting glucose and G6P in the assay are not fully quantitative.

3) Proximity-based biotin-labeling (paragraph with line 126): It identifies proteins at the proximity of the bait, rather than those directly interacting with the bait (only some of them may be directly interact with the bait). The statements “we search for proteins that interact with CpGT1...” and “To identify the most probable interactors of CpGT1” may be reworded, or the technique needs to be explained.

4) Some copyediting issues:

Line 90: spell out full name for smHA in first appearance.

Line 98: ... indicated “that” CpGT1 and CpGT2...

Line 114: ... two rounds of “DNA” (?) replication...

Line 165: between CpGT1 KO strain with CpGT2 (should be CpGT1?) TAG control strain.

Line 190: spell out full name in first appearance (*Escherichia coli*).

Line 214: add gene ID in first appearance (cgd6_2450)

Fig. 1c and 1d: Clarify the color code of the drawing (on top view, the 3HA should not form ring).

Fig. 2b: clarify the color on the plate. Fig. 2d contains two typos (Blotinylated proximal)? Fig. 2e: clarify the “Z-score”.

Line 789: CpGT1-mTurb-3HA should be CpGT1-mTurbo-3HA?

Reviewer #3 (Remarks to the Author):

This study provides evidence that the major hexose transporters in the obligate intracellular stages of *Cryptosporidium parvum* facilitate the salvage of sugar phosphates from intestinal epithelial cells as an essential energy source. *Cryptosporidium* spp cause important diseases in humans and animals and reside within a unique parasitophorous vacuole in intestinal epithelial cells. These protists are predicted to be highly dependent on glycolysis for ATP synthesis, although little is known about how they salvage glucose or other sugars from the host cell. The authors show that obligate intracellular parasite stages up-regulate the expression of the putative glucose transporters, CpGT1 and CpGT2 and localize these transporters to the feeder organelle, an extensive network of membranes that forms an interface between the parasite and the host cell. Genetic deletion of CpGT1 or CpGT2 individually leads to either no, or only a modest, decrease in parasite growth in HCT-8 intestinal cells, respectively, suggesting that these transporters have redundant functions. Unexpectedly, the authors show that loss of hexose kinase (HK) also has no effect on parasite growth, suggesting that CpGT1/2 may be transporting sugar phosphates rather than glucose. This idea was supported from complementation/growth studies on a *E. coli* mutant lacking native glucose transporters, but expressing CpGT1 or CpGT2. Finally, the authors provided evidence using a new conditional knock-down system for Cp, that a putative glycogen phosphorylase is also required for parasite growth, suggestive of a role of amylopectin turnover in maintaining parasite Glc6P levels.

The findings of this study provide new insights into nutrient salvage mechanisms in these important parasites and represents the first example of a sugar phosphate salvage pathway in a eukaryotic/protist pathogen (similar to those found in several intracellular bacterial pathogens). The paper is well written and the data are generally consistent with the proposed conclusions. However, a number of controls are missing and the data also raise other possibilities that could be considered. For example, it is not clear whether disruption of the HK gene results in complete loss of hexose kinase activity due to the presence of other sugar kinases (including pentose kinases). The possibility that CpGT1/2 can transport other sugar phosphates or sugar nucleotides (as has been proposed for *Plasmodium* liver stages) should also be considered/tested. Finally, further studies are needed to confirm the phenotype of the GP mutant and the potential role of amylopectin breakdown in energy homeostasis. These points are outlined in more detail below.

Major points.

The authors show that deletion of the single hexose kinase gene has no effect on intracellular parasite growth. While Cp appears to lack other HK or GlcK homologues, it is possible that these parasites may express other (non-annotated) sugar kinases, including pentose kinases (RibK) that compensate for loss of HK. The conclusion that HK is not essential for parasite growth would therefore be strengthened if the authors could show that lysates from the HK mutant line lack detectable sugar kinase activity.

As neither CpGT1 or CpGT2 are required for parasite growth in HCT-8 cells it remains unclear whether these transporters are functional *in vivo* and/or transport Glc/Glc6P directly. Have the authors considered testing the sensitivity of the GT1/2 mutant lines to glycolytic inhibitors such as 2DG (which should be transported by CpGT1/2 before or after conversion to 2-deoxyglucose-6-P by the host cell).

The *E. coli* complementation experiments provide nice evidence that CpGT1 and 2 can take up Glc6P and Glc6P/Glc, respectively. However, given that CpGT2 can take up glucose, it would be essential to show that the CpGT2-expressing *E. coli* are not secreting/expressing surface phosphatases that could convert Glc6P in the medium to glucose. Have the authors also considered the possibility that CpGT1/2 could be taking up other sugar phosphates that would bypass the requirement for HK, such as Glc1P and UDP-Glc. In this context, it has been proposed, based on essentiality scores, that *P. falciparum* may express transporters for sugar nucleotides.

The authors propose that amylopectin turnover constitutes another mechanism by which Cp intracellular stages generate Glc6P, based on the essentiality of glycogen phosphorylase. However, loss of glycogen phosphorylase could also lead to the toxic accumulation of amylopectin (as is the case when GP is disrupted in *Toxoplasma gondii*), rather than because it has a role in regulating

Glc6P levels in the parasite. This conclusion could be strengthened by imaging of amylopectin granules in GP mutant parasites following conditional knock-down of GP.

It is not conceptually clear how Glc6P transported by CpGT1/2 in the feeder organelle gets to the cytoplasm of the parasite. Is the feeder organelle continuous with the plasma membrane of the parasites, or do CpGT1/2 transport Glc/Glc6P into the lumen of the parasitophorous vacuole and there is a second family of transporters that transport these pools across the parasite PM?

Minor comment.

The authors refer to the Cp glycogen phosphorylase as a debranching enzyme. Debranching enzymes are trans/endoglycosidases that cleave glycosidic linkages within glucan chains. GPs release single Glc1P residues from the non-reducing termini.

Reviewer #1 (Remarks to the Author):

Rui Xu et al. investigated how *C. parvum* obtains vital sugars from its host cells. They figured out two glucose transporters, CpGT1 and CpGT2 localized in the feeder organelle, were not essential for parasite growth through CRISPR-Cas9 strategy. In addition, they analyzed biotinylated proteins by CpGT1-miniTurbo-3HA to identify additional proteins localized to the feeder organelle. Complementation studies in *E. coli* demonstrated the two transporters could uptake G6P. Moreover, they suggested aldolase and glycogen phosphorylase as essential enzymes for parasite growth through CRISPR-Cas9 strategy.

In summary, the manuscript explores metabolic pathways of *C. parvum* with various techniques such as CRISPR-Cas9 knock out/knock in, proximity labeling, complementation studies in *E. coli*, and mAID degron system. Overall, the article seems to contribute to the understanding of *C. parvum* biology. The authors may find the specific points below useful to improve the manuscript.

Suggestions/questions:

1. In several experiments, the authors attached genetic tags such as HA, fluorescent protein, miniTurbo, mAID, etc., to specific proteins. The authors may need to show these genetic tags didn't interfere the specific proteins' functions.

We agree that introduction of tags can sometimes disrupt function, but we think that is unlikely here as tagged lines grow normally, even for essential genes. Unfortunately, we do not have antibodies to the endogenous proteins and generating these reagents would take considerable time with no guarantee that specific sera could be obtained. We have added a statement to the discussion to acknowledge this limitation.

2. In Figure 2e, no biotin condition shows streptavidin signal, while biotin condition doesn't. Please provide correct images.

We apologize that these were inadvertently mislabeled. We have corrected the labels and it now shows there is no streptavidin signal in – biotin but the cells are well-labeled in + biotin.

3. In Figure 2e or 2f, please provide WT CpGT1 (without miniTurbo) +biotin control images.

We have provided images of the wild type parasites (non-tagged CpGT) incubated + biotin to show there is no labeling under these conditions. The images are found in Supplementary Figure 5.

4. In Figure 2f, please provide HA blotting to confirm the enzyme expression levels were same.

Unfortunately, Western blots are very difficult to conduct with transgenic parasites since they require large number of oocysts to detect a positive signal. Since all the parasites express the Turbo fusion, there is good reason to expect that two groups express the same level of enzyme. The only difference here is they are grown in parallel (from the same stock) and one is labeled with biotin while the other is not. As further validation of the similar levels of parasites in the two samples, similar peptide counts were obtained for contaminating proteins in the MS. These are now included Supplementary Table 2. We have added a statement to the text to clarify.

5. Please provide more details on proximity labeling experiment in the methods section. For example, how much cells/proteins were used for SA-beads enrichment? And how much SA-beads were used?

We have added a statement to the text to clarify these details.

6. Have you analyzed biotinylated samples without streptavidin enrichment? Are the bands in -biotin condition endogenously biotinylated proteins in HCT-8 cells?

We have not analyzed samples without enrichment, but the bands seen in the – biotin samples are endogenous proteins. We can determine this because they are among the most abundant host proteins in the MS in both the + and – biotin samples. They correspond to host acetyl-CoA carboxylase, pyruvate carboxylase and propionyl CoA carboxylase. Host proteins are now shown in a separate tab in Supplementary Table 2. We have added a statement to the text to clarify.

7. Could you clarify the reason for incubating the CpGT1-miniTurbo parasites for 19 hr?

We apologize if this was not clear in the manuscript, but the labeling with biotin was done for only one hour. Biotin was added at 19 hours post infection, and the cells harvested at 20 hpi. We have added a statement to the text to clarify.

8. In proteomics dataset, could you provide spectrum intensities, not only number of unique peptides?

We have included the spectral counts in Supplementary Table 2 for both the enriched parasite proteins and host contaminants. They are found in separate tabs as noted by the labels.

9. To map non-restricted organelles, like the feeder organelle, it would be necessary to incorporate spatial controls, such as *C. parvum* cell membrane, for performing ratiometric analysis. The ratiometric analysis would enable the distinction of proteins that are universally abundant or easily biotinylated. For example, ABCC9 could've been labeled easily due to its big size.

We agree that additional imaging studies might help better define the geometry of the feeder organelle in reference to the parasite and host membranes. However, we feel such studies are outside the scope of the present study. Instead, we have added transmission EM images to Figure 1 to help show the arrangement of the parasitophorous vacuole, the feeder organelle and the host interface. We agree that the efficiency of protein labeling might be affected by size (as well as its abundance, sensitivity to trypsin and detection in MS). We have added a statement to the text to reflect that multiple features can affect spectral abundance or peptides detected across different targets, making it difficult to compare.

10. As the authors have developed a conditional knockout system in *C. parvum*, is it possible to apply the same approach to aldolase? Failures of KO strain construction might not be enough evidence to define the enzyme as an essential one.

We agree that failure to disrupt aldolase might be due to technical difficulties in disrupting the locus and so we cannot be sure that the failure to obtain a knockout implies essentiality. It would be possible to test this using a regulated system and we have indicated in the Discussion

that this is an important point for future investigation.

Reviewer #2 (Remarks to the Author):

In this manuscript, Xu et al. reports that the zoonotic protozoon *Cryptosporidium parvum* (Cp) possesses multiple transporters to acquire glucose or phosphorylated glucose from host cells to start the glycolysis. Significant discoveries include:

- 1) the investigators produced transgenic parasite lines that tagged two sugar transporters with 3HA, namely CpGT1 (cgd3_4070) and CpGT2 (cgd4_2870), and confirmed by immunolabeling and immuno-EM that both transporters were located to the “feeder organelle” during the parasite intracellular development, but absent in the motile sporozoites.
- 2) Since the unique “feeder organelle” has been poorly understood, they used a proximity-labeling technique to discover a number of new proteins potentially localized in this organelle (mostly transporters). They further validated that one of these proteins, CpACBP1, was localized in the feeder organelle.
- 3) CpGT1 and CpGT2, as well as the hexokinase (CpHK, cgd6_3800), were individually dispensable, indicating that Cp uses multiple pathways to start the glycolysis (some redundancy in acquisition of sugars). They also implied that glycolysis was essential based on the failure of knockout (KO) of aldolase gene in the parasite.
- 4) The glucan phosphorylase (CpGP, cgd6_2450) was essential (indispensable) for the parasite growth in vivo as implied by the failure of gene KO. Conditional KO suggested the CpGP plays a role in the parasite growth “in vitro”.
- 5) By transformation of *Escherichia coli* mutants deficient with glucose and glucose-6-phosphate (G6P) transporters with parasite transporters for complementary experiments, they showed that CpGT1 may transport both glucose and G6P, while CpGT2 may only transport G6P.

These discoveries are highly significant in understanding the parasite acquisition of carbohydrates from the host for energy metabolism and for carbon source, the function of feeder organelle unique to *Cryptosporidium* parasites. I have no major concerns, but a few minor comments as follows.

- 1) The starch (amylopectin) as a carbon storage could be essential for the environmental and motile stages, such as sporozoites in the oocysts and merozoites, but less likely essential to the intracellularly developing stages that could directly acquire mono- or di-sugars or phosphorylated sugars from the host. This may explain the “delayed” effect of CpGP-KO on the parasite growth in vitro (Fig. 5e), i.e., in 48-h in vitro assay, there is a release of merozoites to invade new host cells, by which CpGP-KO merozoites could only use stored mono/di-sugars and could not stay extracellularly for less time.

We agree that the delay in observing a phenotype for the GP knockout might reflect differential use of sugar sources in early and late stages. We have added a statement to the Discussion to address this possibility.

2) In E. coli complementary assays (Fig. 4), parasite CpGT1 and CpGT2 may not function optimally for many reasons. Therefore, the comparison on the efficiency of CpGT1 between transporting glucose and G6P in the assay are not fully quantitative.

We completely agree that the complementation assays are difficult to compare in a quantitative manner as we cannot be sure the proteins are expressed at similar levels, or may not be folded, or modified in the same way they are in the parasites. We have added a statement to the Discussion to address this deficiency.

3) Proximity-based biotin-labeling (paragraph with line 126): It identifies proteins at the proximity of the bait, rather than those directly interacting with the bait (only some of them may be directly interact with the bait). The statements “we search for proteins that interact with CpGT1...” and “To identify the most probable interactors of CpGT1” may be reworded, or the technique needs to be explained.

This is an excellent point and we have revised the text to more carefully indicate what is being detected in the labeling experiments. More specifically we have changed “interact” to “are in close contact”.

We have corrected these items.

4) Some copyediting issues:

Line 90: spell out full name for smHA in first appearance.

Line 98: ... indicated “that” CpGT1 and CpGT2...

Line 114: ... two rounds of “DNA” (?) replication...

Line 165: between CpGT1 KO strain with CpGT2 (should be CpGT1?) TAG control strain.

Line 190: spell out full name in first appearance (Escherichia coli).

Line 214: add gene ID in first appearance (cgd6_2450)

Fig. 1c and 1d: Clarify the color code of the drawing (on top view, the 3HA should not form ring). We have updated the cartoon image to reflect the protein distribution The nucleus has been removed from 1C and 1D as it is outside the pane of focus.

Fig. 2b: clarify the color on the plate. Fig. 2d contains two typos (Blotinylated proximal)? Fig. 2e: clarify the “Z-score”.

We have added a description to the methods of how these values were generated and plotted.

Line 789: CpGT1-mTurb-3HA should be CpGT1-mTurbo-3HA?

Reviewer #3 (Remarks to the Author):

This study provides evidence that the major hexose transporters in the obligate intracellular stages of *Cryptosporidium parvum* facilitate the salvage of sugar phosphates from intestinal epithelial cells as an essential energy source. *Cryptosporidium* spp cause important diseases in humans and animals and reside within a unique parasitophorous vacuole in intestinal epithelial

cells. These protists are predicted to be highly dependent on glycolysis for ATP synthesis, although little is known about how they salvage glucose or other sugars from the host cell. The authors show that obligate intracellular parasite stages up-regulate the expression of the putative glucose transporters, CpGT1 and CpGT2 and localize these transporters to the feeder organelle, an extensive network of membranes that forms an interface between the parasite and the host cell. Genetic deletion of CpGT1 or CpGT2 individually leads to either no, or only a modest, decrease in parasite growth in HCT-8 intestinal cells, respectively, suggesting that these transporters have redundant functions. Unexpectedly, the authors show that loss of hexose kinase (HK) also has no effect on parasite growth, suggesting that CpGT1/2 may be transporting sugar phosphates rather than glucose. This idea was supported from complementation/growth studies on a *E. coli* mutant lacking native glucose transporters, but expressing CpGT1 or CpGT2. Finally, the authors provided evidence using a new conditional knock-down system for Cp, that a putative glycogen phosphorylase is also required for parasite growth, suggestive of a role of amylopectin turnover in maintaining parasite Glc6P levels.

The findings of this study provide new insights into nutrient salvage mechanisms in these important parasites and represents the first example of a sugar phosphate salvage pathway in a eukaryotic/protist pathogen (similar to those found in several intracellular bacterial pathogens). The paper is well written and the data are generally consistent with the proposed conclusions. However, a number of controls are missing and the data also raise other possibilities that could be considered. For example, it is not clear whether disruption of the HK gene results in complete loss of hexose kinase activity due to the presence of other sugar kinases (including pentose kinases). The possibility that CpGT1/2 can transport other sugar phosphates or sugar nucleotides (as has been proposed for *Plasmodium* liver stages) should also be considered/tested. Finally, further studies are needed to confirm the phenotype of the GP mutant and the potential role of amylopectin breakdown in energy homeostasis. These points are outlined in more detail below.

Major points.

The authors show that deletion of the single hexose kinase gene has no effect on intracellular parasite growth. While Cp appears to lack other HK or GlcK homologues, it is possible that these parasites may express other (non-annotated) sugar kinases, including pentose kinases (RibK) that compensate for loss of HK. The conclusion that HK is not essential for parasite growth would therefore be strengthened if the authors could show that lysates from the HK mutant line lack detectable sugar kinase activity.

This is an interesting possibility, although it could be challenging to show such activity in lysates since necessary cofactors and/or physical conditions for a putative sugar kinase might be hard to anticipate. As such, a negative result would still not be definitive. We acknowledge this is an important consideration for future studies, we have revised the text to account for this alternative possibility.

As neither CpGT1 or CpGT2 are required for parasite growth in HCT-8 cells it remains unclear whether these transporters are functional in vivo and/or transport Glc/Glc6P directly. Have the authors considered testing the sensitivity of the GT1/2 mutant lines to glycolytic inhibitors such

as 2DG (which should be transported by CpGT1/2 before or after conversion to 2-deoxyglucose-6-P by the host cell).

We have tested the growth inhibition with 2DG in wild type parasites and shown they are susceptible (see below). As such, we might expect that the mutants would be less sensitive, either due to lower transport of 2DG or the phosphorylated form. However, since the function of these two transporters are partially overlapping, it is unlikely that either single mutant would have a strong phenotype. A better approach would be to combine a single knockout with a conditional knockdown. Although this should now be feasible in *C. parvum*, it has not yet been implemented. We have acknowledged this limitation and suggested it as a future direction.

The *E. coli* complementation experiments provide nice evidence that CpGT1 and 2 can take up Glc6P and Glc6P/Glc, respectively. However, given that CpGT2 can take up glucose, it would be essential to show that the CpGT2-expressing *E. coli* are not secreting/expressing surface phosphatases that could convert Glc6P in the medium to glucose. Have the authors also considered the possibility that CpGT1/2 could be taking up other sugar phosphates that would by-pass the requirement for HK, such as Glc1P and UDP-Glc. In this context, it has been proposed, based on essentiality scores, that *P. falciparum* may express transporters for sugar nucleotides.

We think it is highly unlikely that a surface phosphate is converting G6P to glucose as this would mask the phenotype of the *E. coli* upht mutant (this strain has three systems for glucose uptake). The possibility that the parasite also transports sugar nucleotides is interesting and we have mentioned this alternative in the Discussion along with a reference summarizing the repertoire of nucleotide sugar transporters in the genome, most of which are unstudied.

The authors propose that amylopectin turnover constitutes another mechanism by which Cp intracellular stages generate Glc6P, based on the essentiality of glycogen phosphorylase. However, loss of glycogen phosphorylase could also lead to the toxic accumulation of amylopectin (as is the case when GP is disrupted in *Toxoplasma gondii*), rather than because it has a role in regulating Glc6P levels in the parasite. This conclusion could be strengthened by imaging of amylopectin granules in GP mutant parasites following conditional knock-down of GP.

This is an interesting point and something we have also considered. We tried hard to image the amylopectin by PAS staining, but the high levels of O-linked sugars in Cp resulted in a very high background that compromises this approach. Instead, we used transmission EM to examine cells during development as it is known that amylopectin accumulates in the macrogamont stage. Following shut down of GP, we observed a striking increase in the number of small, electron lucent vesicles, consistent with deposition of AP. As such, we agree that another reason for the inhibition of growth following loss of HP could be an imbalance in the amount of stored AP. We have revised the text to include this alternative explanation and added images to Figure 5 to demonstrate this phenotype.

It is not conceptually clear how Glc6P transported by CpGT1/2 in the feeder organelle gets to the cytoplasm of the parasite. Is the feeder organelle continuous with the plasma membrane of the parasites, or do CpGT1/2 transport Glc/Glc6P into the lumen of the parasitophorous vacuole and there is a second family of transporters that transport these pools across the parasite PM?

This is an excellent point and we thank the reviewer for bringing it to our attention. We have gone back to the TEM images of the early stages of the FO development to resolve the membranes that surround the parasite and separate it from the host. The parasitophorous vacuole membrane does not extend beneath the parasite, but rather reverses direction and becomes contiguous with the parasite plasma membrane. The FO is continuous with the parasite PM at the base of the parasite. As such, the FO is exposed to the host cytosol and separated from it by a dense cytoskeletal interface. The permeability of this interface has not been tested, but it is likely permeable to small molecules that diffuse to the FO where they are transported into the parasites. We have added images to Figure 1 to verify this arrangement and revised the model in Figure 6 to better illustrate the membranes.

Minor comment.

The authors refer to the Cp glycogen phosphorylase as a debranching enzyme. Debranching enzymes are trans/endoglycosidases that cleave glycosidic linkages within glucan chains. GPs release single Glc1P residues from the non-reducing termini.

Thank you for pointing this out. We have corrected the terminology of the GP enzyme.

REVIEWER COMMENTS

Reviewer #1 (Remarks to the Author):

I thank the authors for addressing most of my comments. The manuscript has been improved; however, I still have remaining questions and concerns as below.

1. In Fig. 1b, there are only 11 transmembrane domains, supposed to be 12. Please correct the figure.

2. As a non-parasite biologist, I noticed that schematic side views in Fig. 1c and 1d don't look similar to the 3D images. In the Fig. 1c side view, Hoechst staining shows multiple nuclei closely located to the feeder organelle. And in the Fig. 1d side view, it seems that anti-HA stained outside of 1E12 staining, not inside of it. Considering that HA is tagged to the cytosolic side of Cp and 1E12 stains the membrane of Cp, it would be more natural if anti-HA staining were inside of 1E12 staining. Can you provide clarification on this? In addition, there is no explanation on 1E12 in the main manuscript.

3. In Fig. 1e and f, it is a bit hard to interpret the images. It would be helpful if you could label the location of the feeder organelle on the images (and if possible, on other EM images as well). Additionally, in Fig. 1f, there are three dots that are not close to the host membrane. Do you think this region could be the feeder organelle? Using APEX+DAB instead of immune-gold EM might have resulted in clearer images of the feeder organelle.

4. In Fig. 2e, the merged image of +Biotin doesn't show Hoechst staining, while No Biotin image does. Please correct the images. And in the same figure, the No Biotin images have a lower background compared to Biotin images when the brightness is increased. It seems like higher contrast was applied to No biotin images to reduce background, or the images were taken at different microscope settings. Could you provide the No biotin images with similar contrast? The reason I increased the brightness of the images was to check for the host mitochondria staining, since the most intense bands in Fig. 2f are biotinylated mitochondrial proteins of the host cells. (It could be invisible if the laser power was low.)

5. In Fig. 2f, it is surprising that the streptavidin blot doesn't show the self-biotinylation band of CpGT1-miniTurbo. Typically, the self-biotinylation band is the most intense band when using promiscuous biotin ligase such as miniTurbo. However, in the blot, ~37kDa band is the most intense one. Since miniTurbo is ~28 kDa and CpGT1 is ~54 kDa (?), it should be > 82 kDa (PTM could increase the size), but there's no intense band in that range. You mentioned that HA blotting is difficult to conduct, but since you used the LICOR system, you could've simply co-stained with anti-HA and anti-rat-680RD. One possibility is that CpGT1-miniTurbo wasn't solubilized well in the lysis buffer you used (1% NP-40 buffer). If biotinylated proteins were not solubilized well in the buffer, it could've affected proteomics result as well.

6. Regarding the proteomics data, omitting biotin is not a proper control for proximity labeling. It would be more accurate to include a cytosolic and/or membrane-targeted miniTurbo that doesn't localize to the feeder organelle. Moreover, I think the amount of materials for LC-MS/MS wasn't sufficient if you used 6-well scale as described in the Methods. And lysis buffer that properly solubilizes target proteins could've improved the result. Overall, the proximity labeling experiment seems to have some issues. However, the other main discoveries of the manuscript are not obtained from the proteomics data, and you have successfully validated CpABC1 is localized to the feeder organelle, so it would be ok to have this data in the manuscript as is. Or it might be better to move this data to the Supplementary figure.

7. In Supplementary Fig. 5, the Merge image is not actually a merged image. Please correct the label or provide a merged image. And it seems HA and Strep images have been swapped. HA image shows red background, while Strep image shows green background at high brightness settings.

8. In Fig 3, 'TAG' are HA-tagged lines, right? As you mentioned that in the response, to

demonstrate that tagged lines grow normally, could you provide data comparing the growth of tagged lines to that of non-tagged lines?

Reviewer #2 (Remarks to the Author):

The authors have satisfactorily addressed and clarified all concerns in the revised manuscript. I have no more comments concerning the overall quality and significance of the study.

Reviewer #3 (Remarks to the Author):

I think the authors have satisfactorily addressed the comments raised in the three reviews, with additional data and changes to the text. I commend them on an important and very interesting study.

REVIEWER COMMENTS

Reviewer #1 (Remarks to the Author):

I thank the authors for addressing most of my comments. The manuscript has been improved; however, I still have remaining questions and concerns as below.

We would like to thank the reviewer for the positive comments and opportunity to address the remaining points. We realize that parasites, especially the understudied organism *Cryptosporidium* used here, will not be familiar to many readers. As such, we welcome the opportunity to provide better explanation for the novel morphological features of the parasite membranes especially the feeder organelle, the development of new methodological approaches in this system, and novel findings and interpretations provided by our study.

Although we can appreciate the rationale for a number of the requests for additional data, acquiring these data turn out to be very in this system due to intrinsic limitations to growing the parasite. To provide some perspective, generating 10 million cells- sufficient for a single Turbo labeling experiment or performing a Western blot requires use of multiple animals to amplify the parasites over a 4 week period of infection – followed by the time required to actually conduct the experiment. As such, many things that are routine in other systems are extremely challenging here. That said, we are pleased to be able to provide the first example of permissive biotin labeling in *Cryptosporidium* and while perhaps not optimal, our results have been validated for several candidates. We are also the first to assign any protein to the feeder organelle, having localized three putative transporters there by confocal and immunoEM. We hope the reviewer will take these challenges into account in evaluating the responses below.

1. In Fig. 1b, there are only 11 transmembrane domains, supposed to be 12. Please correct the figure.

Thank you for pointing out this discrepancy. We have redrawn the figure.

2. As a non-parasite biologist, I noticed that schematic side views in Fig. 1c and 1d don't look similar to the 3D images. In the Fig. 1c side view, Hoechst staining shows multiple nuclei closely located to the feeder organelle. And in the Fig. 1d side view, it seems that anti-HA stained outside of 1E12 staining, not inside of it. Considering that HA is tagged to the cytosolic side of Cp and 1E12 stains the membrane of Cp, it would be more natural if anti-HA staining were inside of 1E12 staining. Can you provide clarification on this? In addition, there is no explanation on 1E12 in the main manuscript.

We apologize for the simplified view shown in the cartons. They were not meant to show the exact features of the cells in the IFA images, but rather generic images of these stages in the life cycle. Unfortunately, we chose the wrong stages for showing the multinucleated meront (lower images 1c, 1d) and we can see why this was confusing. We have updated the figure with a more appropriate cartoon version of immature and mature meronts (Fig 1 c,d). We have also added clarification that 1E12 stains the parasite membrane at all stages. Prior to merozoite segmentation (separation of daughter cells), the pattern of 1E12 outlines the composite multinucleate form. (Fig 1d right). After cell division, it stains the surface of individual

merozoites (Fig 1d left). The reviewer picked up on an important point that we previously overlooked: 1E12 is absent from the feeder organelle- host interface. This is consistent with the fact that neither the parasite surface membrane nor the parasitophorous vacuole membrane extends beneath the parasite. When the mature merozoites divide during merogony, they become separated from a residual cytoplasm that connects to the feeder organelle to the host cell. As such, it is entirely consistent that feeder organelle localized proteins might lie outside of the boundary stained by 1E12. This is also shown in the cartoon versions (the red line in Fig 1 c,d lower panels) where the PVM and parasite membrane do not extend below the parasite, yet FO proteins (green) are exposed at this interface. CpGT1 and CpGT2 have slightly different patterns, which do not depend on the stages of maturation of the meront, as clarified in the text. We have added a description to the Results to better explain the staining features of 1E12 and to highlight this important observation.

3. In Fig. 1e and f, it is a bit hard to interpret the images. It would be helpful if you could label the location of the feeder organelle on the images (and if possible, on other EM images as well). Additionally, in Fig. 1f, there are three dots that are not close to the host membrane. Do you think this region could be the feeder organelle? Using APEX+DAB instead of immune-gold EM might have resulted in clearer images of the feeder organelle.

We can appreciate that the feeder organelle (FO) is a challenging structure for readers not familiar with the parasite, and indeed it is not well described in the literature beyond a few descriptive morphological studies. The FO is an extensive system of membrane invaginations that have been described as dome shaped, circular and/or tubular in structure (Bochimoto et al., 2009 (5)). These membranes lie between the host cell cytosol and the parasite, but they do not comprise a uniform flat interface typical of a normal membrane. Instead, they form convoluted invaginations that extend upward into the parasite cytosol from the host parasite interface as seen in immunoEM (Fig 1e,f) or conventional TEM (Fig 1 g,h). The gold dots that lie more distant from the host cell interface in Fig 1e,f are in fact found along these extended convoluted membranes and are part of the FO. To aid the reader in understanding this complex organelle, we have added a better description of the FO in the Introduction (referencing prior work) and Results (highlighting our findings). We also include brackets to highlight this interface in the EM images. Although APEX-DAB could potentially provide a more refined view of this interface, it would necessitate generating new transgenic lines and then optimizing conditions for detection. As such, we feel such improvements are better left to future studies.

4. In Fig. 2e, the merged image of +Biotin doesn't show Hoechst staining, while No Biotin image does. Please correct the images. And in the same figure, the No Biotin images have a lower background compared to Biotin images when the brightness is increased. It seems like higher contrast was applied to No biotin images to reduce background, or the images were taken at different microscope settings. Could you provide the No biotin images with similar contrast? The reason I increased the brightness of the images was to check for the host mitochondria staining, since the most intense bands in Fig. 2f are biotinylated mitochondrial proteins of the host cells. (It could be invisible if the laser power was low.)

The images were acquired at the same exposure and from the same experiment. After capture, they were adjusted in ImageJ to enhance the brightness in the blue channel as otherwise it can be difficult to identify the parasite nuclei. The other channels were not adjusted between the + and - biotin images. The reason for the apparent higher background in the + Biotin image when the gain is increased substantially is presumably due to the fact that addition of biotin results in

some background binding to how cells and the substrate and this is picked up by the SA detection. This does not happen in the – Biotin sample, resulting in lower background. However, we would like to point out that at the exposure shown here, the backgrounds are comparable. We have included a statement in the Methods describing image adjustment.

5. In Fig. 2f, it is surprising that the streptavidin blot doesn't show the self-biotinylation band of CpGT1-miniTurbo. Typically, the self-biotinylation band is the most intense band when using promiscuous biotin ligase such as miniTurbo. However, in the blot, ~37kDa band is the most intense one. Since miniTurbo is ~28 kDa and CpGT1 is ~54 kDa (?), it should be > 82 kDa (PTM could increase the size), but there's no intense band in that range. You mentioned that HA blotting is difficult to conduct, but since you used the LICOR system, you could've simply co-stained with anti-HA and anti-rat-680RD. One possibility is that CpGT1-miniTurbo wasn't solubilized well in the lysis buffer you used (1% NP-40 buffer). If biotinylated proteins were not solubilized well in the buffer, it could've affected proteomics result as well.

We certainly acknowledge that the SA blot does not show a prominent band at the size of the bait, and this is also consistent with the fact that CpGT1 is not among the major proteins identified in MS analysis. As suggested, this could occur due to less efficient extraction, or differential sensitivity to trypsin or detection in MS. We have not performed the double staining by western blot with anti-HA since we had already validated by IFA that the protein was expressed and correctly localized. Repeating this experiment to confirm by western blot would require use of multiple animals and > 30 days of infection to collect sufficient oocysts. Given the limited additional insight this would provide (confirming expression of a band of the expected size), it does not seem justifiable. Instead, we have added a statement to the Results to highlight the potential limitations with the MS data (see below). Despite these caveats, we feel the MS data are useful for predicting new components of the FO, as demonstrated by the ABC transporter that we validated by epitope tagging, IFA, and immune-EM as shown in Fig S6. Collectively, we feel the data will be useful for the community and would prefer to keep the summary of the TurboID data in the main Figure 2 as currently configured.

Although the bait protein CpGT1 was detected in the dataset, it was not among the most abundant proteins, consistent with the observation that it was not recognized as a major band in the streptavidin blot (Fig. 2e) suggesting it might not be easily solubilized. Additionally, it is difficult to infer the abundance of actual interacting proteins from the frequency of detected peptides since biotin labeling and identification are influenced by protein size, proximity to the bait, available residues for conjugation, sensitivity to trypsin, and detection by MS.

6. Regarding the proteomics data, omitting biotin is not a proper control for proximity labeling. It would be more accurate to include a cytosolic and/or membrane-targeted miniTurbo that doesn't localize to the feeder organelle. Moreover, I think the amount of materials for LC-MS/MS wasn't sufficient if you used 6-well scale as described in the Methods. And lysis buffer that properly solubilizes target proteins could've improved the result. Overall, the proximity labeling experiment seems to have some issues. However, the other main discoveries of the manuscript are not obtained from the proteomics data, and you have successfully validated CpABC1 is

localized to the feeder organelle, so it would be ok to have this data in the manuscript as is. Or it might be better to move this data to the Supplementary figure.

We agree that there are other possible controls, some of which might be considered more rigorous, and there certainly might be better ways to solubilize samples. However, the point of these experiments was not to optimize the protocol but to use it as a discovery step to highlight other potential proteins that are enriched in the feeder organelle. We used an infected 6 well plate simply because it is very difficult to produce the biomass of parasites necessary to perform these experiments and scaling them to larger formats is simply not feasible. Despite this limitation, we were able to detect many proteins that were statistically enriched, including the ABC transporter, whose location at the FO was validated by tagging (Fig S6). We have added a statement to the Discussion to acknowledge these limitations and indicate that future experiments could improve on the method to further validate the composition of the feeder organelle.

Our identification of several proteins that are translocated to the feeder organelle, as well as defining the kinetics for its assembly, will support future studies to define the origin and composition of the membranes, and features that direct proteins to this interface. Additionally, future studies are likely to build on these findings through alternative extraction techniques to improve the efficiency of recovery of membrane proteins and additional controls to rule out contaminants.

7. In Supplementary Fig. 5, the Merge image is not actually a merged image. Please correct the label or provide a merged image. And it seems HA and Strep images have been swapped. HA image shows red background, while Strep image shows green background at high brightness settings.

We apologize for this error. Since the images are essentially blank, we inadvertently mixed them up when exporting to create the composite. We added a new channel for merge and clarified that the previous image panel was rabbit-anti PanCp colored cyan.

8. In Fig 3, 'TAG' are HA-tagged lines, right? As you mentioned that in the response, to demonstrate that tagged lines grow normally, could you provide data comparing the growth of tagged lines to that of non-tagged lines?

The behavior of the tagged lines (Fig 3) is similar to many others we have generated in the past, which is the reason we characterized them as "grow normally". To clarify this statement, we have provided a statement to the Results about the kinetics and peak burden in mice that show expansion that is consistent with the behavior of previously characterized tagged lines. However, a direct comparison to untagged lines is not straight forward as wild type parasites are expanded in calves while tagged lines are generated in mice. As a consequence of expansion in these different hosts, they can have different growth properties when tested in mice. As well, the wild type line is not tagged with luciferase, so the readout would need to be completely different (qPCR for example). For this reason in our previous studies, and again here, we have compared the knockout lines to their respective tagged lines. In this comparison, CgGT1 shows no defect (the TAG line is the same as KO) while CpGT2 KO shows decreased fitness compared to the TAG line.

Reviewer #2 (Remarks to the Author):

The authors have satisfactorily addressed and clarified all concerns in the revised manuscript. I have no more comments concerning the overall quality and significance of the study.

Thank you for your helpful comments.

Reviewer #3 (Remarks to the Author):

I think the authors have satisfactorily addressed the comments raised in the three reviews, with additional data and changes to the text. I commend them on an important and very interesting study.

Thank you for your comments and endorsement of the importance of our studies.